

# Fauna associated with shallow-water methane seeps in the Laptev Sea

Andrey A. Vedenin[1], Valentin N. Kokarev[2,3], Margarita V. Chikina[3], Alexander B. Basin[3], Sergey V. Galkin[4] and Andrey V. Gebruk[4]

[1] Laboratory of plankton communities structure and dynamics, P.P. Shirshov Institute of Oceanology, Moscow, Russia
[2] Faculty of Biosciences and Aquaculture, Nord University, Bodø, Norway
[3] Laboratory of Ecology of Coastal Bottom Communities, P.P. Shirshov Institute of Oceanology, Moscow, Russia
[4] Laboratory of Ocean Bottom Fauna, P.P. Shirshov Institute of Oceanology, Moscow, Russia

Corresponding author
Andrey A. Vedenin,
urasterias@gmail.com

## ABSTRACT

**Background**. Methane seeps support unique benthic ecosystems in the deep sea existing due to chemosynthetic organic matter. In contrast, in shallow waters there is little or no effect of methane seeps on macrofauna. In the present study we focused on the recently described methane discharge area at the northern Laptev Sea shelf. The aim of this work was to describe the shallow-water methane seep macrofauna and to understand whether there are differences in macrobenthic community structure between the methane seep and background areas.

**Methods**. Samples of macrofauna were taken during three expeditions of RV *Akademik Mstislav Keldysh* in 2015, 2017 and 2018 using 0.1 m$^2$ grabs and the Sigsbee trawl. 21 grabs and two trawls in total were taken at two methane seep sites named *Oden* and *C15*, located at depths of 60–70 m. For control, three 0.1 m$^2$ grabs were taken in area without methane seepage.

**Results**. The abundance of macrofauna was higher at methane seep stations compared to non-seep sites. Cluster analysis revealed five station groups corresponding to control area, *Oden* site and *C15* site (the latter represented by three groups). Taxa responsible for differences among the station groups were mostly widespread Arctic species that were more abundant in samples from methane seep sites. However, high densities of symbiotrophic siboglinids *Oligobrachia* sp. were found exclusively at methane seep stations. In addition, several species possibly new to science were found at several methane seep stations, including the gastropod *Frigidalvania* sp. and the polychaete *Ophryotrocha* sp. The fauna at control stations was represented only by well-known and widespread Arctic taxa. Higher habitat heterogeneity of the *C15* site compared to *Oden* was indicated by the higher number of station groups revealed by cluster analysis and higher species richness in *C15* trawl sample. The development of the described communities at the shallow-water methane seeps can be related to pronounced oligotrophic environment on the northern Siberian shelf.

## INTRODUCTION

Methane gas seeping from the seafloor, similar to hydrothermal vents, can support the conditions for unique fauna largely independent of photosynthetic primary production (*Van Dover, 2000*; *Levin, 2005*; *Dando, 2010*). Distinct faunal response at methane seeps (also known as "cold seeps") associated with increase of abundance and biomass and presence of unique taxa absent in background areas has been described from many areas of the ocean (*Gebruk, 2002*; *Baker & German, 2004*; *Levin, 2005*; *Sommer et al., 2009*). Taxa restricted to methane seeps either develop symbiotic relationships with methanotrophic or sulphide-oxidizing bacteria or feed directly on benthic or suspended bacterial matter. In addition, secondary consumers such as predators feeding exclusively on symbiotrophic taxa and grazers may be present (*Dando, 2010*).

In the Arctic Ocean, several methane seep ecosystems have been discovered and investigated. The most studied include the Håkon Mosby mud volcano (HMMV) in the Norwegian Sea (*Gebruk et al., 2003*) and several sites around Svalbard and at Vestnesa Ridge (*Åström et al., 2016*; *Åström et al., 2018*). Other described cold seeps include the Lofoten-Vesterålen continental margin area (*Sen et al., 2019a*) and mud volcanoes in the Beaufort Sea (*Paull et al., 2015*). The cold seeps inhabited by chemosymbiotic benthic taxa are mostly located below the photic zone (depth >200 m both around Svalbard and at HMVV) (*Gebruk et al., 2003*; *Åström et al., 2016*). At the same time, macrobenthic communities from areas with extensive methane discharge located at shallow depths (e.g., in the Norwegian and White Seas at depths <100 m) show no or minor response in terms of chemosymbiotic communities development (*Savvichev et al., 2004*; *Levin, 2005*). However, at the shallow-water cold seeps certain changes in abundance/biomass of the common macrofaunal taxa are often observed, caused by enhanced vertical mixing, productivity and additional hard substrata (carbonate crusts) due to extensive methane discharge (*Jensen et al., 1992*; *Sahling et al., 2003*; *Pohlman et al., 2017*). In general, a depth boundary is observed between shallow-water cold seeps and their "deep-sea" counterparts at approximately 200 m (*Tarasov et al., 2005*; *Dando, 2010*). One of possible reasons for this boundary is the origin of organic matter: at depths <200 m photosynthetic organic matter is more available for benthic consumers due to stronger bentho-pelagic coupling (*Levin, 2005*; *Dando, 2010*), while at greater depth the amount of photosynthetic organic matter decreases and chemosynthesis starts to play a significant role for local organic matter production. Therefore, despite the presence of methane and sulfides (unfavorable for most organisms due to toxicity), unique and diverse ecosystems can develop at deep-sea cold seeps (*Powell, Bright & Brooks, 1986*; *Dando et al., 1993*; *Dando, 2010*).

Fauna associated with cold seeps in the Arctic includes symbiotrophic siboglinid polychaetes and thyasirid bivalves, but mainly consists of widespread Arctic species not restricted to methane seeps (*Gebruk et al., 2003*; *Åström et al., 2016*; *Åström, Oliver & Carroll, 2017*). Arctic cold seep assemblages are characterized by the dominance of frenulate siboglinid worms, while large chemosymbiotrophic methane seep taxa (vestimentiferan worms, bathymodioline and vesicomyid bivalves) are absent (*Sen et al., 2018*). One of the common effects of Arctic methane seeps on macrobenthic communities is an increased

abundance and biomass of regular allochthonous taxa compared to the background (e.g., *Rybakova et al., 2013*; *Åström et al., 2016*), although species richness at cold seeps is usually not higher than in the background (in terms of macrofauna) all over the world (*Levin, 2005*; *Dando, 2010*). However, recent results obtained from the southwestern Barents Sea showed increased taxonomic richness within the seepage sites (*Sen et al., 2019b*). Outside the Arctic similar species richness patterns were reported from the Gulf of Mexico (*Cordes et al., 2010*) and Argentina (*Bravo et al., 2018*).

In the Siberian Arctic, areas of intense methane discharge (methane seeps) were discovered on the outer shelf of the Laptev Sea in 2008 (*Yusupov et al., 2010*). Further research revealed numerous gas flares in the northern Laptev Sea shelf (*Lobkovsky et al., 2015*; *Shakhova et al., 2015*). Within this area specific microbial communities based on methane oxidation were discovered (*Savvichev et al., 2018*). *Baranov et al. (in press)* suggested that methane seeps occur through the fault system belonging to the Laptev Sea Rift system and the Khatanga-Lomonosov Fracture Zone located between the Eurasian and North American Tectonic Plates. The faults may conduct the gas from reservoirs deep in the sediment below the caprock formed by permafrost and gas hydrates (*Shakhova et al., 2015*; *Thornton et al., 2016*; *Baranov et al., in press*). Within the seeping area, multiple bacterial mats and occasional methane bubbles and carbonate crusts were observed (*Baranov et al., in press*).

The methane associated fauna was registered on the Laptev Sea shelf and slope much earlier. During expeditions of RV *Polarstern* in 1993 and 1995 five species of siboglinids were found in this area in the depth range 50–2,000 m (*Sirenko et al., 2004*), which is more species than anywhere else in the high Arctic (*Buzhinskaja, 2010*).

We examined benthic communities associated with methane seeps in the Laptev Sea at two sites: *C15*, centred around 76°47.4′N and 125°49.5′E with depths 70–73 m and *Oden*, centred around 76.894°N and 127.798°E, with depths 63–67 m. Preliminary description of benthic megafauna based on video recordings obtained by ROV was published by *Baranov et al. (in press)*. The aim of this study is to describe the biological peculiarities of the methane seep macrofauna based on grab and trawl samples and to reveal differences in either integral community characteristics or distribution of certain species between the methane seep and background areas. We hypothesized that the seep sites are different from the non-seep in terms of macrofauna (general community characteristics and certain species distribution).

## MATERIALS & METHODS

Samples of macrofauna were taken during three expeditions of RV *Akademik Mstislav Keldysh* in 2015 (AMK-63), 2017 (AMK-69) and 2018 (AMK-72) on the northern Laptev Sea shelf, in the area of active methane discharge. The gears used for sampling included the *Okean* (in 2015) and *van Veen* (in 2017–2018) grabs (0.1 m$^2$ sampling area) and the *Sigsbee* trawl (2 m frame width) (*Eleftheriou & McIntyre, 2005*). 21 grab and two trawl stations were performed in total at three sites: on two methane seep fields (12 grabs and one trawl at *C15*, 70.6–73.0 m depth, and six grabs and one trawl at *Oden*, 63.0–63.1 m depth) and

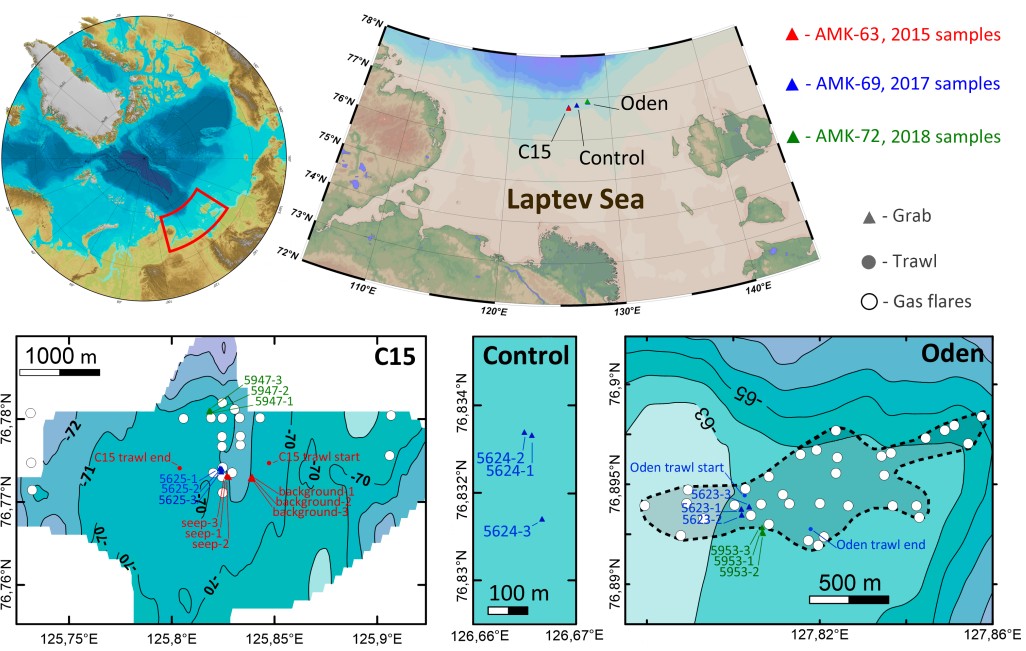

**Figure 1** **Study area.** Enlarged maps show sampling sites and corresponding stations. Detailed bathymetry is only available for *C15* and *Oden* sites; white circles indicate previously recorded gas flares (*Baranov et al., in press*). The dotted line at *Oden* site enclosed map shows the approximate perimeter of seeping area.

at the control site with no methane seeping (3 grabs, 69.6–69.7 m depth) (Fig. 1). A single trawl was taken at each seep site to minimize the possible ecosystem damage from this gear. In 2015 three seep stations were performed at sites where gas bubbling (gas flares) was visible on echo-sounder. Three more grabs were taken ~200 m away from the nearest gas flare to catch background community. In 2017 and 2018 station selection was based largely on the previously mapped methane flares (*Baranov et al., in press*). All the 2017 and 2018 grabs were taken above the gas flares (Fig. 1). Station data with coordinates and depths are shown in Table 1. For additional information on methane seep fields see *Flint et al. (2018)* and *Baranov et al. (in press)*.

Sediment from grab samples was washed by hand through the 0.5 mm mesh size sieve, and fixed with buffered 4% formalin solution afterwards. Two grab samples from the expedition in 2018 (Stat. 5947-3 at *C15* and 5953-2 at *Oden* site) were fixed with 96% ethanol. A 10-litre subsample of sediment taken from each trawl catch was washed through the one mm mesh size sieve and then fixed with neutralized 4% formalin. The material obtained was analyzed in the laboratory; all macrofaunal organisms were identified to the lowest possible taxonomical level and counted. Species from grab samples were weighed (wet weight, all specimens of each species at a time). Mollusks were weighed with shells, polychaetes with calcareous (spirorbids) or mucous tubes (*Spiochaetopterus typicus* and siboglinids) were weighed with tubes. Density and biomass were calculated per square meter for grab samples. Dominant species were distinguished by biomass. For trawl samples we

**Table 1 Data on stations used in the present study. For trawl stations coordinates and depth of start and end are given.**

| Expedition, year | Station | Site | Gear | Latitude | Longitude | Depth (m) |
|---|---|---|---|---|---|---|
| **AMK-63, 2015** | seep-1 | C15 | Okean-0.1 | 76°46.376′N | 125°49.641′E | 72.0 |
| | seep-2 | C15 | Okean-0.1 | 76°46.376′N | 125°49.664′E | 72.3 |
| | seep-3 | C15 | Okean-0.1 | 76°46.379′N | 125°49.618′E | 72.4 |
| | background-1 | C15 | Okean-0.1 | 76°46.375′N | 125°50.346′E | 73.0 |
| | background-2 | C15 | Okean-0.1 | 76°46.366′N | 125°50.366′E | 73.0 |
| | background-3 | C15 | Okean-0.1 | 76°46.365′N | 125°50.339′E | 73.0 |
| | C15 trawl | C15 | Sigsbee | 76°46.483′N | 125°50.843′E | 71.5 |
| | | | | 76°46.447′N | 125°48.231′E | 72.0 |
| **AMK-69, 2017** | 5623-1 | Oden | Van Veen-0.1 | 76°53.624′N | 127°48.110′E | 63.0 |
| | 5623-2 | Oden | Van Veen-0.1 | 76°53.608′N | 127°48.114′E | 63.1 |
| | 5623-3 | Oden | Van Veen-0.1 | 76°53.632′N | 127°48.219′E | 63.0 |
| | Oden trawl | Oden | Sigsbee | 76°53,667′N | 127°48,157′E | 63.0 |
| | | | | 76°53,566′N | 127°49,075′E | 63.0 |
| | 5624-1 | Control | Van Veen-0.1 | 76°49.998′N | 126°39.936′E | 69.6 |
| | 5624-2 | Control | Van Veen-0.1 | 76°50.003′N | 126°39.896′E | 69.7 |
| | 5624-3 | Control | Van Veen-0.1 | 76°49.883′N | 126°40.000′E | 69.6 |
| | 5625-1 | C15 | Van Veen-0.1 | 76°46.438′N | 125°49.417′E | 70.8 |
| | 5625-2 | C15 | Van Veen-0.1 | 76°46.435′N | 125°49.442′E | 70.7 |
| | 5625-3 | C15 | Van Veen-0.1 | 76°46.413′N | 125°49.437′E | 70.6 |
| **AMK-72, 2018** | 5947-1 | C15 | Van Veen-0.1 | 76°46.847′N | 125°49.085′E | 72.3 |
| | 5947-2 | C15 | Van Veen-0.1 | 76°46.847′N | 125°49.085′E | 72.0 |
| | 5947-3 | C15 | Van Veen-0.1 | 76°46.848′N | 125°49.097′E | 72.0 |
| | 5953-1 | Oden | Van Veen-0.1 | 76°53.554′N | 127°48.405′E | 63.0 |
| | 5953-2 | Oden | Van Veen-0.1 | 76°53.551′N | 127°48.409′E | 63.0 |
| | 5953-3 | Oden | Van Veen-0.1 | 76°53.567′N | 127°48.400′E | 63.0 |

calculated the contribution (in %) of each species to abundance. Biomass was not measured for trawl samples due to poor state of preservation. For ethanol fixed samples from Stat. 5947-3 and 5053-2, the biomass loss was corrected using taxa-specific coefficients after *Brotskaya & Zenkevich (1939)*.

For grab samples total abundance, biomass, species richness (species number), Pielou evenness, Hurlbert rarefaction index and Shannon-Wiener diversity index (H' ln) were calculated to get integral community characteristics. Abundance and biomass data from grab samples were standardized and square root transformed to increase the role of rare taxa and to reduce the impact of highly abundant taxa. The similarity among grab samples was estimated using the quantitative index of Bray-Curtis. Clusters were built based on similarity matrices using the unconstrained tree routine (UNCTREE); results were verified by SIMPROF to distinguish different station groups with significant differences in species composition. The results from cluster analysis were verified by non-metric multidimensional scaling (n-MDS). Clusters revealed by these methods were defined as separate station groups in terms of quantitative taxonomical similarity. Shade plots were built to visualize the species abundance and biomass differences between the stations

and species in clusters. The Kruskal-Wallis test was used to verify differences in certain taxa distribution among station groups, followed by Dunn's post-hoc test (with Holm adjustment for multiple comparisons). Species-individuals accumulation curves were plotted for each station group (*McCune, Grace & Urban, 2002*; *Clarke & Gorley, 2015*; *R Development Core Team, 2020*).

For all species present in any station group, an algorithm estimating the likelihood of accidental catch was applied. If a uniform distribution of species between two sampling efforts A and B is assumed, the probability of species absence at each station of B-sampling would be $(1- P_A)^{N(B)}$, where N(B) is the number of stations in B-sampling and $P_A$ is the species occurrence (the proportion of stations where the species was present) in A-sampling. Using this equation, the likelihood of accidental absence of any species in either station group can be estimated. The number of grabs required for species catch in B-sampling can be calculated by the equation: $n = \lg(\alpha)\lg(1- P_A)^{-1}$, where $\alpha$ is the likelihood of species finding in B-sampling taken as 0.99 (*Azovsky, 2018*; *Vedenin et al., 2019*).

For trawl samples, the species rank distributions were plotted. Species richness, Pielou evenness, Hurlbert rarefaction index and Shannon-Wiener diversity index were calculated using the taxa-percentage values. Differences between trawl catches were estimated by similarity percentage routine (SIMPER).

Statistical analyses were performed in Primer V6, V7, Past 3.0 and R V3.6.1 with R package Dunn.test V1.3.5 software (*Clarke & Warwick, 2001*; *Hammer, 2013*; *Clarke & Gorley, 2015*; *R Development Core Team, 2020*).

## RESULTS

A total of 289 taxa of benthic macrofauna were identified in grab and trawl samples. In grab samples, density varied from 580 ind. $m^{-2}$ (St. 5624-3, Control site) to 9880 ind. $m^{-2}$ (St. seep-3, *C15* site). Biomass ranged from 16.28 g ww $m^{-2}$ (St. seep-1, *C15* site) to 405.79 g ww $m^{-2}$ (St. 5623-3, *Oden* site). The list of all identified taxa from trawl and grab samples, with values of abundance and biomass is given in the Supplemental Information 1.

### Grab samples

Unconstrained tree with SIMPROF analysis revealed five significantly distinct groups of samples (Fig. 2). The UNCTREE parameters are shown in Supplemental Information 2. The groups partly corresponded with the station locations and presence/absence of methane seeps (*Control*, *C15* and *Oden* sites). To avoid a mix-up between the station groups and seeping sites hereinafter the corresponding names are used with either –station group or –site ending.

Square root transformed biomass data are used. Samples and knots that were not significantly different at *p* <0.05 are connected with red dashed lines. Green lines indicate SIMPROF groups.

### Characteristics of station groups

At the *Control* station group (Fig. 1), the bivalve *Portlandia arctica* comprised most of the biomass at all three stations, followed by the starfish *Ctenodiscus crispatus* and the bivalve

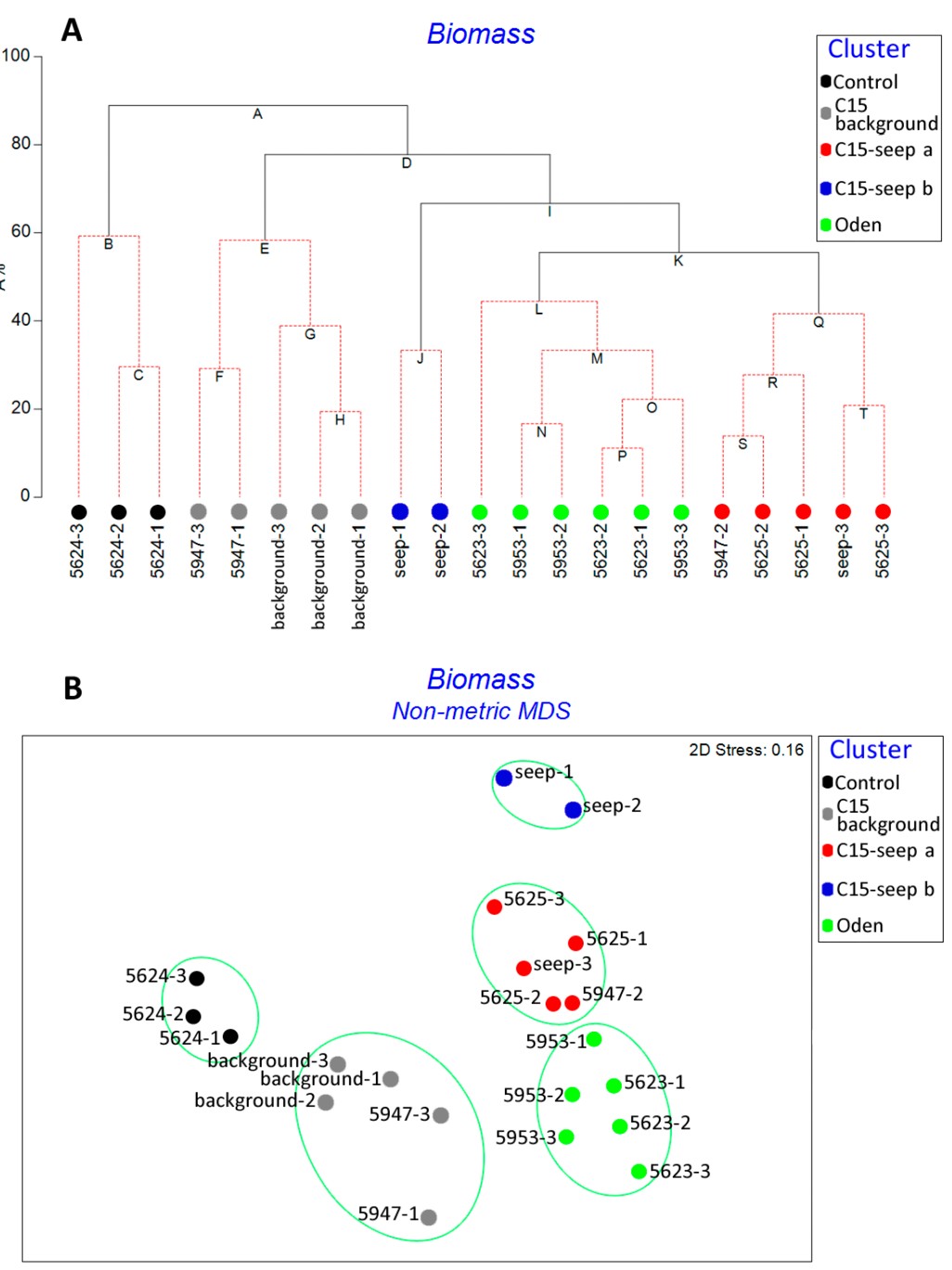

**Figure 2** **UNCTREE analysis with SIMPROF results (A) and non-metric multidimensional scaling plot (B) of grab stations using the Bray-Curtis similarity index.** Square root transformed biomass data are used. Samples and knots that were not significantly different at $p < 0.05$ are connected with red dashed lines. Green lines indicate SIMPROF groups.

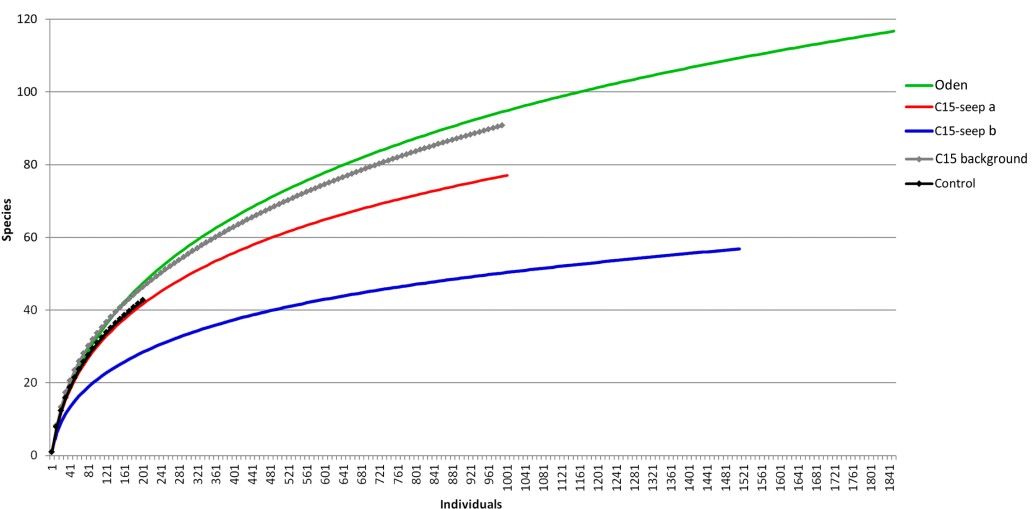

**Figure 3** **The species-individuals accumulation curves for the station groups.** Colors are the same as in Fig. 2.

*Macoma calcarea*. Due to the low number of samples, the species-individuals accumulation curve did not reach the saturation point (Fig. 3). *Control* station group had in comparison to the other groups from *C15* and *Oden* seep sites the lowest density and species richness, whereas the evenness was the highest (Fig. 4).

The *C15-seep a* station group included five stations, all within the *C15* seep site. In this group, biomass and diversity values were intermediate among other station groups. Dominant species in this group were the bivalve *Nuculana pernula*, the siboglinid *Oligobrachia* sp. and the polychaete *Cistenides hyperborea*. Species-individuals accumulation curve in this group reached saturation due to the largest number of samples (Fig. 3).

The *C15-seep b* station group consisted of only two stations from *C15* site. This group demonstrated the highest abundance values and low biomass among all groups. Dominant species included small polychaetes *Cossura longocirrata, Micronephthys minuta* and *Ophryotrocha* sp. (Fig. 4, Supplemental Information 1).

The *Oden* station group included six stations, all located within the *Oden* seep site. Values of biomass, species richness and diversity indices in this group were the highest among all station groups (Fig. 4). The main dominant species were the siboglinid *Oligobrachia* sp. and the polychaetes *Myriochele heeri* and *Nephtys ciliata*.

The last group *C15 background* contained five stations taken within the *C15* site. Taxonomical composition at these stations was similar to that in the *Control* group, with the bivalve *Portlandia arctica* being the dominant species. Bivalves *Yoldiella lenticula* and *Y. solidula* were subdominant. In this station group, the biomass values were the lowest, other general community characteristics were intermediate (Fig. 4). Similar to the *Control* group, *C15 background* did not reach the saturation point at species-individuals accumulation plot (Fig. 3).

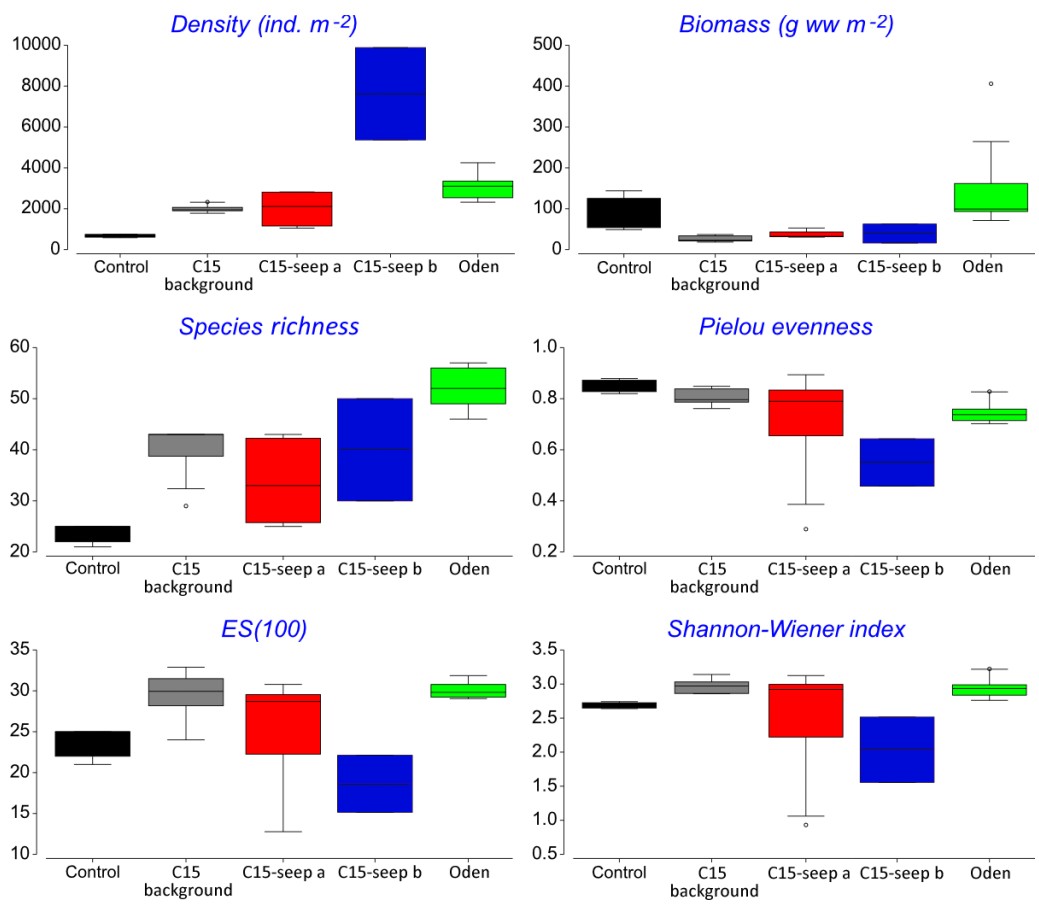

**Figure 4    Univariative characteristics of identified clusters expressed in standard box plots.** Values of total density, biomass, species richness, Pielou evenness, Hurlbert rarefaction index and Shannon-Wiener index are shown. Each graph contains interquartile ranges (colored boxes), mean values (horizontal line inside each box) and minimum and maximum values (lines outside the boxes). Exact values of these characteristics are shown in Supplemental Information 3. Colors are the same as in Figs. 2 and 3.

## Comparison of seep and non-seep station groups

General community characteristics in the station groups appeared different in abundance, biomass and diversity (Fig. 4, Supplemental Information 3). The abundance of several taxa varied significantly among five station groups (*Control*, *C15 background*, *C15 seep a*, *C15 seep b* and *Oden*) (Fig. 5). The Kruskal-Wallis test showed that differences in abundance of at least ten species are statistically reliable (Table 2). Statistics of the Dunn's test are shown in Supplemental Information 4. The seep sites were characterized by higher densities of the polychaetes *Tharyx* sp. and *Cistenides hyperborea* and the ophiuroid *Ophiocten sericeum*. On the contrary, the bivalve *Portlandia arctica* was markedly more abundant in *Control* and, to a lesser extent, in *C15 background* station groups (Fig. 5). Notable were extreme densities of small polychaetes at some seep stations, including *Cossura longocirrata* and *Ophtyotrocha* sp. (Fig. 5A) at *C15 seep b*.

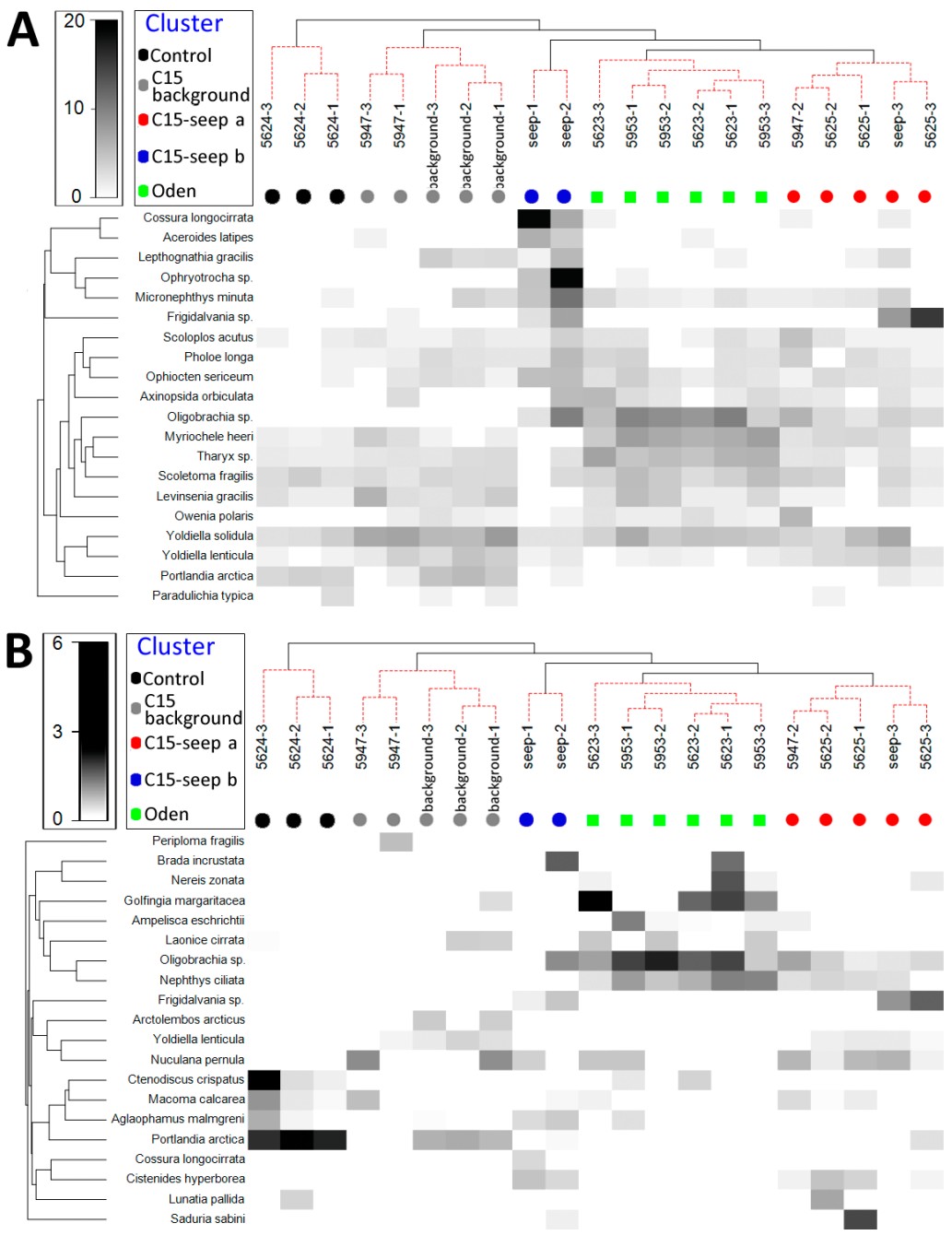

**Figure 5  Shade plot of species square root transformed abundance (A) and biomass (B) at stations arranged by clusters.** The species list is reduced to 20 most important taxa. Order of stations and colors the same as in Fig. 2. Taxa grouped in clusters using UPGMA algorithm based on index of association.

**Table 2 Results of the Kruskal-Wallis and Dunn's post-hoc tests for taxa with different abundance values in five station groups.** Mean abundance in each station group is shown. Taxa are arranged according to $p$-value. Pairs in Dunn's pairwise comparisons column indicate significant comparisons (Dunn's $p < \alpha/2 = 0.025$).

| Species | Mean abundance in station groups | | | | | Kruskal-Wallis | | Dunn's pairwise comparisons |
|---|---|---|---|---|---|---|---|---|
| | 1 | 2 | 3 | 4 | 5 | $H$ (chi$^2$) | $p$ | |
| *Oligobrachia* sp. | 0 | 0 | 63.0 | 14.6 | 53.5 | 16.80 | 0 | 1–3; 2–3 |
| *Nephtys ciliata* | 0 | 0 | 2.2 | 1.6 | 0 | 15.72 | 0 | 2–3 |
| *Yoldiella lenticula* | 0.7 | 12.6 | 1.0 | 8.8 | 2.0 | 15.59 | 0 | 2–3 |
| *Cistenides hyperborea* | 0 | 0.4 | 0.2 | 4.4 | 5.5 | 13.49 | 0 | 3–4 |
| *Tharyx* sp. | 2.3 | 6.0 | 32.5 | 3.8 | 2.5 | 14.81 | 0.01 | 1–3; 3–4 |
| *Spiochaetopterus typicus* | 0 | 0 | 3.0 | 0.6 | 0 | 12.95 | 0.01 | 2–3 |
| *Yoldiella solidula* | 9.3 | 44.0 | 23.5 | 19.6 | 5.0 | 12.51 | 0.01 | 2–5 |
| *Cossura longocirrata* | 0 | 0 | 0.2 | 0.6 | 199.0 | 11.47 | 0.02 | 1–5; 2–5; 3–5 |
| *Ophiocten sericeum* | 0.3 | 3.8 | 6.3 | 4.8 | 26.5 | 10.52 | 0.03 | 1–5 |
| *Portlandia arctica* | 0 | 14.6 | 0 | 0.8 | 0.5 | 10.52 | 0.03 | 2–3 |
| *Pleusymtes pulchella* | 0 | 0 | 0.5 | 1.2 | 3.5 | 10.37 | 0.03 | no values |
| *Frigidalvania* sp. | 0 | 0 | 0 | 59.8 | 29.5 | 9.23 | 0.06 | no values |
| *Anobothrus gracilis* | 0 | 0 | 1.2 | 0.2 | 0 | 5.06 | 0.06 | no values |
| *Axinopsida orbiculata* | 0 | 1.6 | 9.2 | 2.8 | 14.5 | 7.14 | 0.13 | no values |
| *Paroediceros lynceus* | 0 | 0 | 0 | 0.8 | 4.0 | 6.87 | 0.14 | no values |
| *Haploops tubicola* | 0.3 | 1.4 | 0.2 | 0 | 0 | 6.87 | 0.14 | no values |

**Notes.**
1 –*Control* group; 2 –*C15 background* group; 3 –*Oden* group; 4 –*C15-seep a* group; 5 –*C15-seep-b* group.

Certain species present at some methane seep sites were completely absent at the non-seep sites (Fig. 5). Among them, at least five species (the polychaetes *Spiochaetopterus typicus* and *Nephtys ciliata*, the siboglinid *Oligobrachia* sp., the bivalve *Axinopsida orbiculata* and the amphipod *Pleusymtes pulchellus*) were present only at *C15* and *Oden* sites not randomly. At least one species, the undescribed gastropod *Frigidalvania* sp., was present only in *Oden* station group and absent in other station groups not randomly (Table 3). The estimated number of grabs required to catch the latter species was slightly lower than the number of grabs taken.

## Trawl samples

The overall Bray-Curtis similarity between the two trawls was 65.6%. Species ranking graphs showed high level of dominance by abundance for both trawl stations (Fig. 6). The dominant species in both trawls was the ophiuroid *Ophiocten sericeum*: 37% of the total abundance at *C15* and 46% at *Oden*. The second most abundant species at *C15* was the gastropod *Frigidalvania* sp. (12%) and at *Oden* the bivalve *Yoldiella solidula* (11%). Ten most abundant species accounted for >70% of the total abundance in both trawls (Fig. 6).

Species richness, Pielou evenness, Hurlbert rarefaction for 100 individuals and Shannon-Wiener index are shown in Table 4. The evenness and ES (100) was higher in the *Oden*-trawl than in the *C15*-trawl, similar to results based on grab samples. However, the species

**Table 3  Species occurrence, likelihood of not finding a species, number of grabs taken and number of grabs required for finding a species calculated for the species present only at seep sites and only at *C15* site.**

| Species | Species occurrence | Likelihood of not finding | Number of grabs | |
|---|---|---|---|---|
| | | | Required for finding ($\alpha = 0.99$) | Taken |
| Species present at *C15-seep a*, *C15-seep b* and *Oden* and absent at *C15 background* and *Control* sites | | | | |
| *Spiochaetopterus typicus*[a] | 0.62 | 4.79E−04 | 4.8 | 8 |
| *Nephtys ciliata* | 0.77 | 8.04E−06 | 3.1 | 8 |
| *Cossura longocirrata* | 0.38 | 0.021 | 9.5 | 8 |
| *Anobothrus gracilis* | 0.38 | 0.021 | 9.5 | 8 |
| *Oligobrachia* sp.[a] | 1 | 0 | 1 | 8 |
| *Axinopsida orbiculata*[a] | 0.77 | 8.04E−06 | 3.1 | 8 |
| *Paroediceros lynceus* | 0.23 | 0.123 | 17.6 | 8 |
| *Pleusymtes pulchella*[a] | 0.53 | 0.002 | 6.0 | 8 |
| Species present at *Oden* and absent at *C15-seep a* and *C15-seep b* | | | | |
| *Frigidalvania* sp.[a] | 0.57 | 0.006 | 5.4 | 6 |
| *Portlandia arctica* | 0.43 | 0.035 | 8.2 | 6 |
| *Paroediceros lynceus* | 0.43 | 0.035 | 8.2 | 6 |

**Notes.**
[a] Species absent not accidentally.

richness (as well as the total amount of individuals) in the *C15*-trawl was higher than in the *Oden*-trawl (Table 4, Supplemental Information 1).

Species responsible for taxonomical difference between the two trawl samples are shown in Table 5. Most notable is a high abundance of the gastropod *Frigidalvania* sp. at *C15*. At *Oden Frigidalvania* sp. was also present, but in much smaller densities (only 2.3% of the total abundance). In addition, *C15*-sample differs from *Oden* by high numbers of various filter-feeders including 6 species of sponges (with *Craniella polyura* being most numerous), at least 6 species of cnidarians, 17 species of bryozoans and 3 species of tunicates (Table 6).

At *C15* trawl sample, a large piece of carbonate crust was found. Cavities of its pores were inhabited by numerous polychaetes, also recognized from the soft sediments around the seepage area (e.g., members of families Nephthyidae, Nereididae, Oweniidae and Terebellidae, see Supplemental Information 1), and by several filter-feeders (Hydrozoa).

## Comparison of C15 and Oden sites

All gears showed significant differences between the *C15* and *Oden* sites expressed in different taxonomical composition and quantitative characteristics. The Bray-Curtis similarity between the sites according to the grab samples and trawl samples was 26.2 and 65.6, respectively. The main differences in species composition included the high abundance of the sponge *Craniella polyura* and the gastropod *Frigidalvania* sp. at *C15* site and higher numbers of the ophiuroid *Ophiocten sericeum* at *Oden* site.

The grab samples indicated a high level of heterogeneity in benthic fauna distribution. Some species formed patches, for example *Oligobrachia* sp., *Cossura longocirrata* and *Ophryotrocha* sp., being extremely numerous at some and moderate in abundance at the

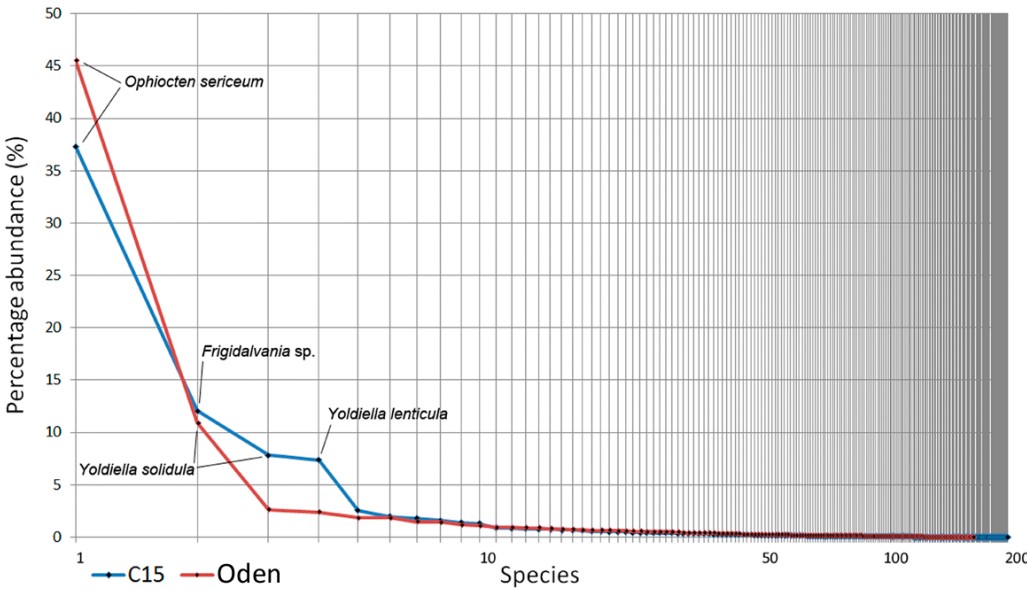

**Figure 6** **Species ranking for *C15.* and *Oden* trawl samples.** The most numerous species are indicated. X-axis is logarithmic.

**Table 4** **Species richness, Pielou evenness, Hurlbert rarefaction for 100 individuals and Shannon-Wiener index calculated for trawl samples.**

| Trawl | Species richness | Pielou evenness | ES (100) | Shannon-Wiener index |
|---|---|---|---|---|
| *C15* | 203 | 0.55 | 29.97 | 2.92 |
| *Oden* | 167 | 0.56 | 33.02 | 2.86 |

neighboring grab stations. There were also species with rather uniform distribution based on combined data, for example *Ophiocten sericeum*. According to the cluster analysis, the *C15* site is more heterogenic forming at least three different species complexes within its area (Fig. 2). Dissimilarities within the *C15* and *Oden* sites were 64.7 and 26, respectively (Supplemental Information 2).

Dominant species were different in grab and trawl samples. The dominant species in trawls at the two methane seep sites was the ophiuroid *Ophiocten sericeum*, whereas for grab samples, the siboglinid *Oligobrachia* sp., the bivalve *Nuculana pernula* and the polychaete *Myriochele heeri* were the most abundant.

## DISCUSSION

### Integral community parameters: methane seep vs. non-seep

The abundance of macrofauna was higher at the seep stations compared to the background, based on the grab samples. In addition, at *Oden* seep site the biomass was higher compared to non-seep sites. High abundance and biomass has been reported from both hydrothermal vents and cold seeps all over the world, compared to the surrounding areas (*Levin, 2005*; *Dando, 2010*). In the Arctic, a twofold increase of biomass compared to control sites was

**Table 5  Similarity percentage routine for trawl samples.** Species with contribution >0.5% are shown.

| Species | Abundance (%) | | Average dissimilarity | Contribution, % | Cumulative, % |
|---|---|---|---|---|---|
| | *C15* | *Oden* | | | |
| *Frigidalvania* **sp.** | **12.05** | 2.37 | 4.84 | 14.06 | 14.06 |
| *Ophiocten sericeum* | 37.32 | 45.55 | 4.11 | 11.94 | 26.00 |
| *Yoldiella lenticula* | **7.37** | 1.11 | 3.13 | 9.08 | 35.08 |
| *Yoldiella solidula* | 7.82 | 10.89 | 1.54 | 4.46 | 39.55 |
| *Portlandia arctica* | **2.56** | 0.15 | 1.21 | 3.51 | 43.05 |
| *Laona finmarchica* | **1.60** | 0.00 | 0.80 | 2.32 | 45.38 |
| *Phascolion strombus* | **1.95** | 0.36 | 0.80 | 2.31 | 47.69 |
| *Myriochele heeri* | 0.42 | 1.82 | 0.70 | 2.03 | 49.72 |
| *Micronephthys minuta* | 0.16 | 1.47 | 0.65 | 1.90 | 51.62 |
| *Craniella polyura* | **1.30** | 0.00 | 0.65 | 1.88 | 53.50 |
| *Pholoe longa* | 1.38 | 2.62 | 0.62 | 1.79 | 55.29 |
| *Munnopsis typica* | 0.67 | 1.84 | 0.59 | 1.70 | 56.99 |
| *Scoletoma fragilis* | 0.28 | 1.17 | 0.45 | 1.30 | 58.30 |
| *Paraoediceros lynceus* | **1.80** | 0.92 | 0.44 | 1.28 | 59.58 |
| *Rostroculodes hanseni* | 0.00 | 0.88 | 0.44 | 1.28 | 60.85 |
| *Nothria hyperborea* | 0.10 | 0.90 | 0.40 | 1.16 | 62.01 |
| *Solariella obscura* | **0.90** | 0.13 | 0.39 | 1.13 | 63.14 |
| *Tharyx* sp. | 0.04 | 0.67 | 0.31 | 0.91 | 64.05 |
| *Axinopsida orbiculata* | 0.00 | 0.61 | 0.30 | 0.88 | 64.93 |
| *Brada villosa* | **0.66** | 0.06 | 0.30 | 0.86 | 65.79 |
| *Arrhis phyllonyx* | 0.25 | 0.82 | 0.28 | 0.83 | 66.62 |
| *Terebellides* aff. *stroemii* | 0.86 | 1.42 | 0.28 | 0.82 | 67.44 |
| *Similipecten greenlandicus* | **0.68** | 0.15 | 0.27 | 0.78 | 68.22 |
| *Cylichna occulta* | **0.74** | 0.21 | 0.27 | 0.77 | 68.99 |
| *Yoldiella frigida* | 0.15 | 0.63 | 0.24 | 0.70 | 69.69 |
| *Cossura longocirrata* | 0.04 | 0.52 | 0.24 | 0.70 | 70.39 |
| *Sabinea septemcarinata* | 0.09 | 0.57 | 0.24 | 0.69 | 71.08 |
| *Nymphon hirtipes* | 0.10 | 0.57 | 0.23 | 0.67 | 71.76 |
| *Cuspidaria glacialis* | **0.80** | 0.38 | 0.21 | 0.61 | 72.37 |
| *Brada incrustata* | 0.04 | 0.46 | 0.21 | 0.61 | 72.98 |
| *Lepidepecreum umbo* | **0.42** | 0.02 | 0.20 | 0.58 | 73.56 |
| *Ephesiella abyssorum* | 0.01 | 0.40 | 0.19 | 0.56 | 74.12 |
| *Nuculana pernula* | 0.54 | 0.92 | 0.19 | 0.56 | 74.67 |
| *Rozinante fragilis* | **0.51** | 0.15 | 0.18 | 0.53 | 75.20 |
| *Philine lima* | 0.00 | 0.36 | 0.18 | 0.52 | 75.72 |
| *Pleusymtes pulchellus* | 0.36 | 0.71 | 0.17 | 0.51 | 76.22 |
| *Owenia polaris* | 0.07 | 0.42 | 0.17 | 0.50 | 76.72 |

**Notes.**
Species more abundant at *C15* are marked with bold.

**Table 6 List of filter-feeding taxa found in trawl samples.**

| Species | Abundance (%) | |
| --- | --- | --- |
| | **C15** | **ODEN** |
| *Porifera* | | |
| *Sycon* sp. | 0.13 | 0.06 |
| *Lycopodina lycopodium* | 0.01 | – |
| *Mycale* sp. | 0.01 | – |
| *Suberites domuncula* | 0.16 | – |
| *Tentorium semisuberites* | 0.01 | – |
| *Craniella polyura* | 1.30 | – |
| Cnidaria | | |
| *Gersemia fruticosa* | 0.03 | – |
| *Gersemia rubiformis* | 0.01 | – |
| *Lucernaria bathyphila* | 0.01 | – |
| *Lafoea dumosa* | 0.06 | 0.15 |
| *Stegopoma plicatile* | 0.01 | – |
| Hydrozoa gen.sp. | 0.01 | – |
| Polychaeta (Sabellidae + Spirorbidae) | | |
| *Branchiomma arcticum* | – | 0.02 |
| *Euchone analis* | – | 0.04 |
| *Euchone papillosa* | 0.17 | 0.02 |
| *Chone duneri* | 0.03 | 0.13 |
| *Bushiella kofiadii* | 0.01 | – |
| *Circeis spirillum* | 0.04 | – |
| Bryozoa | | |
| *Alcyonidium disciforme* | 0.12 | – |
| *Alcyonidium gelatinosum* | 0.09 | – |
| *Crisia eburneodenticulata* | 0.09 | 0.02 |
| *Defrancia lucernaria* | 0.03 | – |
| *Tubulipora fruticosa* | 0.03 | – |
| *Lichenopora* sp. | 0.01 | – |
| *Carbasea carbasea* | 0.04 | – |
| *Eucratea loricata* | 0.12 | – |
| *Tricellaria gracilis* | 0.01 | – |
| *Dendrobeania fruticosa* | 0.04 | – |
| *Kinetoskias smitti* | 0.06 | – |
| *Porella fragilis* | 0.01 | – |
| *Cheilopora sincera* | 0.12 | – |
| *Parasmittina jeffreysi* | 0.03 | – |
| *Pseudoflustra birulai* | 0.03 | – |
| *Pseudoflustra solida* | 0.01 | – |
| *Ramphostomella bilaminata* | 0.03 | – |
| Tunicata | | |
| *Didemnum albidum* | 0.01 | 0.02 |
| *Ascidia* sp. | 0.01 | – |
| *Synoicum pulmonaria* | 0.01 | – |

observed at cold seeps south off Svalbard (mean values of 20.7 vs. 9.8 g ww m$^{-2}$), the abundance increase was less prominent (770 vs. 590 ind. m$^{-2}$) (*Åström et al., 2016*). At the Vestnesa Ridge the infaunal abundance and biomass was almost five times higher compared to a nearby control area (497 vs. 140 ind. m$^{-2}$ and 2.97 vs. 0.48 g ww m$^{-2}$, respectively) (*Åström et al., 2018*). For the Håkon Mosby mud volcano, which is very deep compared to the Laptev Sea cold seeps (1250 vs. 70 m, respectively), though similar in species composition (*Oligobrachia* fields near the seepage zones and *Ophiocten*-dominated background community) the comparison of abundance and biomass with the background is not available. In our study, the abundance at the methane seep sites *C15* and *Oden* was more than four times higher than at the control. However, differences in biomass although pronounced were not statistically significant. Increased biomass at seep habitats is commonly explained by enhanced organic matter supply and habitat heterogeneity (*Gebruk et al., 2003*; *Sen et al., 2018*).

Pielou's evenness was distinctly higher at the *Control* and *C15 background* station groups, which reflects the increased dominance of certain species at seep stations compared to non-seep. Many authors reported high abundance and biomass values of one to few dominant species at various cold seeps (*Gebruk et al., 2003*; *Åström et al., 2016*; *Åström et al., 2018*). This can be caused by conditions less favorable for some background species (due to the presence of methane and sulphides and lower oxygen level), but more favorable for symbiotrophs or grazers (*Powell, Bright & Brooks, 1986*; *Dando et al., 1993*; *Dando, 2010*).

The cold seeps all over the world oceans usually demonstrate lower diversity values (ES(100) and Shannon-Wiener index) compared to the background areas (summarized by *Levin, 2005*). However, species list from grab and trawl samples showed high ES(100) and Shannon-Wiener index values although only two trawls were sampled. Studies on the Siberian shelf using the same gear under the same conditions obtained less than 150 species per trawl (*Galkin & Vedenin, 2015*; *Vedenin, Galkin & Kozlovskiy, 2015*), while a total of 203 species were found in a single *C15* sample. The unusually high diversity may reflect a higher amount of microniches within the *C15* site. This is indirectly confirmed by lower similarity values observed between all *C15* grab samples. Higher habitat heterogeneity at seep sites can increase the overall diversity of benthic fauna (*Bergquist et al., 2003*; *Gebruk et al., 2003*; *Levin, 2005*; *Menot et al., 2010*; *Åström et al., 2018*). The scale of heterogeneity is hard to assess, but based on stations coordinates and the fact that stations 5947-1 and 5947-2 from *C15*-site were grouped in *C15 background* (though planned as seep stations near the gas flares), while station 5947-2 was grouped as *C15-seep a*, we can assume that the scale is less than 5 m (distance between these stations) (Fig. 1; Table 1).

In addition, the diversity values (ES(100) and Shannon-Wiener index) at the *Oden* station group were significantly higher than at the *C15*-site. The reasons for this are unknown so far, since no environmental parameters measured directly at benthic stations are available. The peculiarly higher values of macrofaunal diversity (Shannon-Wiener index) within the cold seeps are known only for a few seep areas, e.g., for the Vestnesa Ridge (*Åström et al., 2018*), for the South-Western Barents Sea (*Sen et al., 2019b*) and for the Bahía Blanca estuary in Argentina (*Bravo et al., 2018*).

## Common shelf taxa responsible for differences in station groups

The station groups revealed by UNKTREE and n-MDS analysis largely corresponded to the geographical position of the *C15*, *Oden* and control sites. Twelve common species widely distributed across the Siberian shelf (Table 2, *Sirenko, 2001*) were largely responsible for increased integral community parameters in our study. Among such species (based on grab samples) were the polychaetes *Spiochaetopterus typicus*, *Cossura longocirrata* and *Tharyx* sp., the bivalve *Macoma calcarea*, the amphipod *Pleusymtes pulchella* and the ophiuroid *Ophiocten sericeum*. In addition, based on trawl data, the sponge *Craniella polyura* was present in high densities at the *C15* site, together with other filter-feeders including cnidarians and bryozoans (Table 6). Apparently the same species aggregations were visible on the video reported by *Baranov et al. (in press)*. All these above-mentioned species have been previously reported from a wide range of areas of the Laptev Sea and adjacent regions (*Sirenko et al., 2004*).

The increased density of common taxa at deep-sea hydrothermal vents and cold seeps is a well-known phenomenon usually explained by increased availability of organic matter in these habitats (*Hessler & Kaharl, 1995*; *Levin, 2005*). In the Arctic, the increased biomass and abundance of common allochthonous species were reported for the Håkon Mosby mud volcano (*Rybakova et al., 2013*), Svalbard (*Åström et al., 2016*) and Vestnesa Ridge cold seeps (*Åström et al., 2018*). Also, a significant increase of abundance of filter feeders (especially sponges) was shown for the Aurora Seamount on the Gakkel Ridge, the only investigated hydrothermal vent in the Central Arctic Ocean (*Boetius, 2015*; *Bünz et al., 2020*). Soft corals and crinoids were reported to be abundant around the vent fields on the Mohn's Ridge (*Sweetman et al., 2013*).

## Taxa specific for methane seep sites

In our study, the most characteristic species from methane seep sites was the siboglinid *Oligobrachia* sp. (Fig. 7A). This species was present at all seep station and absent at all background/control station. This species is morphologically very close to *Oligobrachia haakonmosbiensis* originally described from the Håkon Mosby mud volcano from the depth of ~1200 m (*Smirnov, 2000*). Colonies of *O. haakonmosbiensis* with the biomass reaching 350 g ww m$^{-2}$ were reported from this area (*Gebruk et al., 2003*). Recent phylogenetical analyses showed that the species from the Laptev Sea belongs to a separate, undescribed species of *Oligobrachia* (*Sen et al., 2018*). In the Laptev Sea, *Oligobrachia* sp. is known from different localities, seep and noon-seep, occurring in a wide depth range 100–2,166 m (*Buzhinskaja, 2010*). Our record at 63 m is the shallowest for this species, with high population density and biomass: >1000 ind. m$^{-2}$ and 45 g ww m$^{-2}$ at Stat. 2623-1 and 5953-2 (*Oden* site). Several specimens from 2015-samples (identified as *O. haakonmosbiensis*) were investigated using transmission electron microscopy (*Savvichev et al., 2018*). Usually the endosymbionts of siboglinids are represented by sulphide-oxidizing bacteria (*Rodrigues et al., 2011*; *Lee et al., 2019*), however in the study of *Savvichev et al. (2018)*, the methanotrophic bacteria were found inside the trophosome.

Additionally to the siboglinids, several samples from the seep-sites were also characterized by high abundance of mollusk species. The gastropod *Frigidalvania* sp.

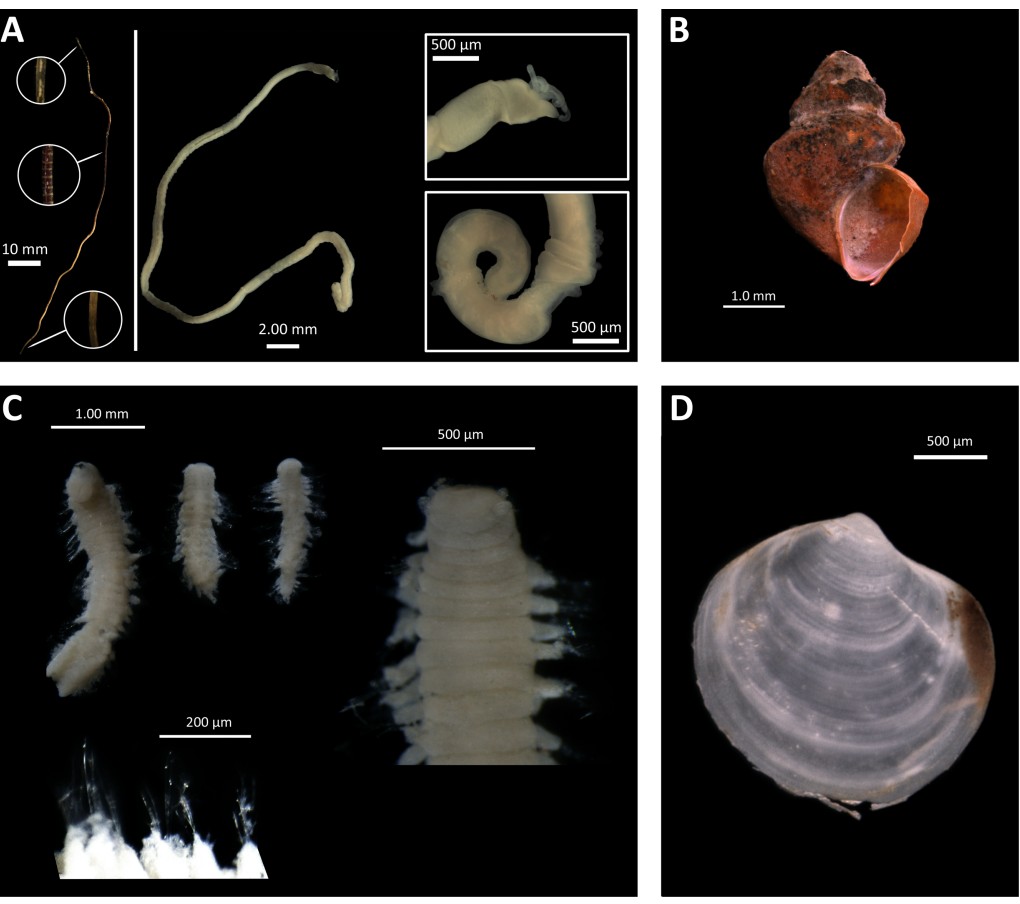

**Figure 7** **Taxa found only at seep stations.** (A), *Oligobrachia* sp. (left—tube with several fragments enlarged; center—complete specimen extracted from tube; right—anterior and posterior fragments of the specimen); (B), *Frigidalvania* sp.; (C), *Ophryotrocha* sp. (upper left—several specimens, total view; upper right—anterior fragment; lower—enlarged parapodia); (D), *Axinopsida orbiculata*. Photos by A. Vedenin and V. Kokarev.

(Rissoidae) occurred in high density at *C15* site: up to 2340 ind. m$^{-2}$ and 25 g ww m$^{-2}$ at St. 5625-3 (Fig. 7B). According to trawl samples, this species occurs at the *Oden* site, but in very low numbers (only 2.4% from total abundance, see Table 5). This species is presumably new to science, based on the presence of a single spiral rib and the absence of knobs, that distinguish it from all known Arctic *Frigidalvania* species (*Warén, 1974*). Subfossils of similar unidentified *Frigidalvania* were found by *Thomsen et al. (2019)* in the sediment cores at the Vestnesa Ridge cold seeps. Large numbers of unknown rissoid gastropods were previously reported from the Håkon Mosby mud volcano, referred to as *Alvania* sp. in *Gebruk et al. (2003)*. The stable isotope analysis showed that the rissoids from the HMMV actively graze on bacterial mats (*Decker & Olu, 2012*). Furthermore, another rissoid gastropod, *Pseudosetia griegi*, was observed grazing on bacterial mats at the hot vent Loki Castle on the Mohn's Ridge (*Sweetman et al., 2013*). At the recently investigated Lofoten canyon seep site dense aggregations of unidentified rissoids were

observed on photographs from ROV (*Sen et al., 2019a*). The details available from the published photo in *Sen et al. (2019a)* (see Fig. 4B) allow us to suggest that the gastropods are likely to belong to genus *Frigidalvania*, based on the shell shape and rusty-brownish periostracum. Unfortunately, in our study we were not able to identify the behavior or lifestyle of *Frigidalvania* sp. This species remained unnoticed in the video data due to its small size (*Baranov et al., in press*). However, multiple bacterial mats observed from video-transects and caught by box corer provide an opportunity for such species to graze on them (*Savvichev et al., 2018*; *Baranov et al., in press*). Other species common at the seep sites in this study, however missing in the background/control was the thyasirid bivalve *Axinopsida orbiculata* (Fig. 7D). Some species of thyasirids are known as symbiotrophic, however, the information on symbiotic bacteria in the gills of *A. orbiculata* is controversial: *Zhukova, Kharlamenko & Gebruk (in press)* have demonstrated the presence of bacteria in bivalve specimens from the Kraternaya Bight, the Kuril Islands, whereas according to *Dufour (2005)* this species lacks bacterial symbionts. It is possible that the high densitiy of *A. orbiculata* is attributed to increased food availability at seep sites. Overall, no bivalves restricted to cold seeps are known so far in the Arctic with the exception of two large thyasirids recently described based on few empty shells (*Åström, Oliver & Carroll, 2017*) and Pleistocene subfossils (e.g., *Archivesica* spp., *Sirenko et al., 2004*; *Hansen et al., 2017*) found deep in the sediment at formerly and/or presently active cold seeps and hydrothermal vents. The subfossils suggest that the Arctic cold seeps (and possibly hydrothermal vents) could be inhabited by richer fauna during late Pleistocene that became extinct during or after Quaternary glaciation (Discussed in *Kim et al., 2006*; *Hansen et al., 2017*; *Thomsen et al., 2019*).

Notable is the high density (>3600 ind. m$^{-2}$) of the dorvilleid polychaete *Ophryotrocha* sp. in one grab sample at *C15-seep b* station group (Supplemental Information 1 (Fig. 7C)). At least 15 species of *Ophtyotrocha* have been described from reducing habitats (*Taboada et al., 2013*; *Salvo et al., 2014*; *Ravara et al., 2015*), including two species considered as obligate for cold seeps in the Kagoshima Bay, Japan (*Miura, 1997*). On the other hand, many species of this genus not restricted to reducing habitats are common in the Arctic seas (*Sirenko, 2001*).

Another taxon common in reducing habitats is tanaid crustaceans (Tanaidacea) (*Larsen, 2006*; *Błazewicz-Paszkowycz & Bamber, 2011*; *Zeppilli et al., 2011*). In our material three species were present (Supplemental Information 1), all widely distributed in the Arctic (*Sirenko, 2001*). The density of tanaids in our samples was low, although this taxon was reported in high densities from the Håkon Mosby mud volcano (*Gebruk et al., 2003*) and the Vestnesa Ridge (*Åström et al., 2018*) with several species (described as new) restricted to the methane seep habitats (*Błazewicz-Paszkowycz & Bamber, 2011*). It seems likely that many species of tanaids remain unidentified and diversity in this taxon remains underestimated owing to difficulties of identification of these small crustaceans (summarized by *Błazewicz-Paszkowycz & Bamber, 2011*). The low number of tanaids in our samples could be a result of a too large sieve mesh size used onboard (see Materials & Methods). Tanaids commonly are <0.5 mm in size and require a corresponding mesh size to be found (*Pavithran et al., 2009*).

Overall, considering grab and trawl data combined, all the seep-specific taxa were the same at both seep sites. The only exception is the polychaete *Ophryotrocha* sp., which could be missed from the *Oden* trawl sample due to the large sieve mesh size (Supplemental Information 1).

## Presence of specific benthic communities at C15 and Oden

Up to now no distinct macrofaunal changes in response to methane seeps were reported from the Arctic Ocean at depths <80 m. In general, at depths <200 m both hydrothermal vents and cold-seeps are usually colonized by a subset of the local fauna (*Tarasov et al., 2005*; *Dando, 2010*). Some species notable at shallow-water methane seeps belong to opportunistic taxa common in various reducing habitats which include siboglinid polychaetes and thyasirid bivalves reported from Skagerrak, Kattegat, coastal areas of Florida, Japan, New Zealand, New Guinea etc. (*Southward & Culter, 1986*; *Schmaljohann & Flügel, 1987*; *Schmaljohann et al., 1990*; *Malakhov, Obzhirov & Tarasov, 1992*; *Gebruk, 2002*). The stable isotope data of nitrogen and carbon sources from various cold seeps all over the ocean suggest that food sources of macrofauna at shallow-water methane seeps are largely photosynthesis-based (*Southward et al., 1996*; *Levin et al., 2000*; *Dando, 2001*; *Levin, 2005*; *Dando, 2010*). It was suggested that the faunistic depth boundary between the deep-sea and shallow-water vents and seeps at approximately 200 m is controlled by the amount of organic matter input from the photosynthetic production (decreasing below the photic zone) and the greater number of predators at shallow depths (summarized by *Dando, 2010*). Seep-obligate species were not reported from depths <200 m (*Tarasov et al., 2005*; *Dando, 2010*).

At the same time, methane seep habitats even at shallow depths increase a number of microniches owing to increased organic matter availability, variety of substrates (including different types of sediment and carbonate crusts) and repeated disturbance (*Sahling et al., 2003*; *Dando, 2010*; *Pohlman et al., 2017*; *Åström et al., 2019*). Therefore, shallow cold-seeps may support greater species diversity (in terms of both richness and diversity indices) compared to the background. In our study at two methane seep sites, *C15* and *Oden*, integral community characteristics were significantly different from those in non-seep areas partially due to presence of species obligate for reducing habitats. In addition, the communities found at *C15* site formed several station groups and were more scattered at the n-MDS plot (Figs. 2A, 2B) which could indicate a larger diversity of microniches within this site. Large amount of filter-feeders (Hydrozoa and Bryozoa) found in *C15*-trawl indicates the presence of hard substrata (including carbonate crusts). The larger amount of microniches is partly supported by the video-data, where the landscape within the active seepages was more complex than in non-seep areas (*Flint et al., 2018*; *Baranov et al., in press*).

Overall, the impact of carbon of different origin and the methane-induced habitat heterogeneity are difficult to separate unless the detailed measurements of environmental parameters and isotope ratios from different organisms are obtained. The isotope data are currently available only for the siboglinid *Oligobrachia* sp., indicating the leading role of

microbial methane oxidation in the nutrition of these symbiotrophic worms (*Savvichev et al., 2018*).

Recent studies conducted at several seep sites south of Svalbard (~300 m depth) showed a certain input of chemosynthesis-based carbon into the largely photosynthesis-based local benthic food web (*Åström et al., 2019*). As an example, the diet of several heterotrophic species, also present in our samples (e.g., polychaetes *Nephtys* sp. and *Scoletoma fragilis*) consisted of chemosynthesis-derived organic at the level of 30–40% (*Åström et al., 2019*). Isotope-based nutritional investigations of shallow-water cold seeps in the North Sea, Northern California and Kattegat showed little or no chemosynthetic contribution to macrofaunal diet (*Dando et al., 1991*; *Jensen et al., 1992*; *Levin et al., 2000*). Based on the increase of density of many species at the seep sites compared to the background (Table 2), it can be assumed that the chemosynthesis-based organic may contribute to the diet of heterotrophic species, such as Nephtyidae polychaetes and other taxa.

Unfortunately, no environmental data except for the echo-sounding showing gas emissions and CTD-measurements obtained from the area of the seeps from only two stations (both away from the benthic sampling sites) are available (*Flint et al., 2018*; *Baranov et al., in press*). Nevertheless, we suggest that response of macrofauna to methane seepage at shallow depths 60–70 m can be related to very low primary productivity on the outer shelf of the Laptev Sea, dropping from ~720 mg C m$^{-2}$ per day at Lena river delta to <100 mg C m$^{-2}$ per day at 600 km (*Sorokin & Sorokin, 1996*) during September. Outside short Arctic summer, these values decrease to almost zero. In these extremely oligotrophic conditions, methane as a source of energy for the methane-oxidizing bacteria stimulates the development of local patchy benthic communities even at a shallow depth. As a comparison, the specific communities with siboglinids around Svalbard located at similar latitude are developed only at depths >200 m, whereas at 80 m no response of macrofauna is observed (*Åström et al., 2016*). Unlike the Laptev Sea shelf, the primary production south and west off Svalbard reaches much higher values up to 1,800 mg C m$^{-2}$ per day during May blooms (*Wassmann et al., 2006*). Furthermore, the Barents Sea remains uncovered with ice during most of the year, while the Laptev Sea shelf is ice-free during one to two months annually. Other examples of shallow-water cold seeps with little or no macrofaunal response are also located in non-oligotrophic areas, such as the North Sea pockmark at 150 m depth with pelagic production up to 1,300 mg C m$^{-2}$ per day (*Moll, 1998*; *Dando et al., 1991*) and the Baltic Sea pockmarks with >2,000 mg C m$^{-2}$ per day in April (*Lignell, 1990*; *Pimenov et al., 2008*).

The only investigated shallow-water reducing habitat comparable in the primary production level to the Laptev Sea is the hydrothermal vent of the volcanic Deception Island, Antarctica. Environmental conditions in the inner bay (Port Foster) of the Deception Island are oligotrophic, most of the surface is ice-covered during austral winter (*Bright et al., 2003*; *Angulo-Preckler et al., 2017*). However, several fumaroles and hot springs in Port Foster, located from intertidal to ~15 m depth enrich the surrounding waters with sulfides and methane and keep the sea surface above them ice-free (*Tilbrook & Karl, 1993*). Certain increase of local macrofauna around the hot springs was described recently (*Angulo-Preckler et al., 2017*). In addition, one taxon, apparently restricted to reducing habitat, was found in

the sediment around the hydrothermal vents. The new taxon was a monocelid flat worm, covered with chemoautotrophic bacteria (*Bright et al., 2003*). However, no seep-specific macrobenthic communities were observed in Port Foster (*Angulo-Preckler et al., 2017*), presumably owing to the very shallow depth: the maximum chemoautotrophic activity and the peak abundance of monocelid worms were recorded at 1–2 m depth (*Bright et al., 2003*).

To confirm the relationship between the depth where the seep-specific communities can develop and the primary production, more shallow-water cold seeps at the extremely oligotrophic conditions (e.g., polar areas) should be investigated.

## CONCLUSIONS

Our study is the first description of shallow-water methane seep communities in the Siberian Arctic. On the northern Laptev Sea shelf, significant differences were found among two methane seep sites (*C15* and *Oden*, located at depths of 63–73 m) and the background areas. The differences included integral community parameters and presence at seep sites of species typical for reducing habitats, such as siboglinids *Oligobrachia* sp. and thyasirid bivalves. Several species at methane seeps are presumably new to science, including the gastropod *Frigidalvania* sp. and the polychaete *Ophryotrocha* sp., found in large densities at *C15* site. We suggest that the distinct influence of methane discharges on macrofauna is more prominent in oligotrophic settings such as the outer shelf of the Laptev Sea.

## ACKNOWLEDGEMENTS

The authors would like to thank the Captain, crew members and shipboard parties of the RV *Akademik Mstislav Keldysh* for multiple help with the work onboard during the 63, 69 and 72 expeditions. Our special thanks to Dr. Michael Flint for organizing the expeditions and for informative discussions. We also thank Dr. Alexey Udalov and Dr. Elena Krylova for help in identifying mollusks and crustaceans.

### Funding

This work was funded by RFBR Grants 18-04-00206, 18-05-60053, 18-05-60228 and the State assignment of IORAS (theme No 0149-2019-0009). The funders had no role in study design, data collection and analysis, decision to publish, or preparation of the manuscript.

### Grant Disclosures

The following grant information was disclosed by the authors:
RFBR: 18-04-00206, 18-05-60053, 18-05-60228.
The State assignment of IORAS (theme No 0149-2019-0009).

### Competing Interests

The authors declare there are no competing interests.

## Author Contributions

- Andrey A. Vedenin and Valentin N. Kokarev conceived and designed the experiments, performed the experiments, analyzed the data, prepared figures and/or tables, authored or reviewed drafts of the paper, and approved the final draft.
- Margarita V. Chikina, Alexander B. Basin and Andrey V. Gebruk analyzed the data, authored or reviewed drafts of the paper, and approved the final draft.
- Sergey V. Galkin conceived and designed the experiments, performed the experiments, analyzed the data, authored or reviewed drafts of the paper, and approved the final draft.

## Data Availability

The raw measurements are available in the Supplemental Files.

## Supplemental Information

Supplemental information for this article can be found online at http://dx.doi.org/10.7717/peerj.9018#supplemental-information.

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
