# Peer review of "Fauna associated with shallow-water methane seeps in the Laptev Sea"

_PeerJ, doi:10.7717/peerj.9018_

## Round 0.1 · original submission · Major Revisions

All referees feel your paper should be published but it needs to be placed in the wider context, the writing needs considerable improvements, and they suggest many aspects that can be improved.

·

Basic reporting

Language:
The English requires editing. Aside from general, grammatical errors, one thing the authors should note is that certain terms or words are used in ways that don’t entirely make sense. This might be a translation issue. I have highlighted some of these instances.

Introduction and background:

The authors set the stage well, starting with an introduction to methane cold seeps and continuing with the main thrust of the manuscript: seeps in shallow waters, particularly in the Arctic. Overall, this approach is good and straightaway makes the research questions of the study clear. However, the language needs to be improved overall. I have included suggestions and comments for refining the introduction below, but this is not a comprehensive list, particularly with respect to the grammar.

Lines 36-37: This first sentence is a little clunky- methane gas itself does not provide environments or habitats. And do you mean unique fauna whose nutrition is derived directly or indirectly, largely independent of photosynthetic primary production?

Line 38: distinct response of benthos/benthic community as expressed as xxx?

Line 41 and 42: you first say Arctic Ocean, and then say HMMV is in the Norwegian Sea. I completely understand the difficulty with the terms, but this terminology is confusing and appears contradictory. Perhaps you can modify it to simply ‘the Arctic’ in line 41, and refer to HMMV being in Nordic waters? Also, you should probably include references to seep sites in the Barents Sea and the Vestnesa site in lines 42-43 as you mention the most studied Arctic seep sites.

Line 45: the way this sentence is currently structured, it seems like the text in the parentheses refers to the photic zones being at 200 and 1200 m.

Line 48: ‘reaction of macrofauna’ is not appropriate language here. It is not the reactions or the responses of the animals/fauna per se that you are examining. Rather, it is how the benthic community is altered by the presence of methane seepage. Please change this throughout.

Line 50: what depth boundary? Ie, what is the boundary of approx. 200 m a boundary for, with respect to CBEs in shallow vs deep locations?

Line 52-53: modify after bivalves to ‘and overall mainly consists of species not unique to methane seeps.’ It might be helpful to include the aggregating part of the sentence as a separate sentence (e.g., this subset of regular/background Arctic benthic fauna tend to aggregate around sites of methane seepage…)

Lines 58-60: This sentence is supposed to be a general statement about seeps, and not specifically about Arctic seeps. In that case, seeps tend to have higher biomass but lower diversity compared to the background overall, not higher diversity as stated here. It is true that in the Arctic though, this trend has been questioned (Gebruk et al., 2002, Sen et al., 2018 Biogeosciences) and even shown to not be true (Åström et al., Limnology and Oceanography, Sen et al., 2019 PeerJ). Please specify and clarify.

Either in in the introduction where gas flares in the Laptev Sea are discussed (briefly), or in the Methods section, some additional detail on the two examined seep sites would be helpful. The figures with the ‘maps’ of the sites and their sampling are white squares and only low resolution bathymetry from Jakobssen et al./IBCAO is used for showing the site locations overall. Some bathymetric maps would be very useful, particularly with locations of flares marked. Is there any information at all on the sources of the methane? In such a shallow setting, this would be useful to know, since gas hydrates are probably unlikely..?

Line 75: change to ‘preliminary descriptions of bottom fauna observed in video are described …..’

Figures:

Figure 1: As stated above, bathymetry maps for the enlarged maps of the three sites would be useful, particularly with gas flares marked on them. This is especially critical since in line 89-90, it says that in 2 years of sampling, station selection was based on where flares were mapped.

Instead of using cruise numbers (AMK 63, 69, etc.), I suggest using years instead.
Instead of a circum-Arctic map at the bottom, it might be more helpful to have a smaller map, showing for example, Russia, or just the Russian/European side of the Arctic. Since it would be possible to show more detail and zoom in more in a map with a smaller extent, the locations of the study sites could then be marked on the map itself, instead of an inset with low resolution bathymetry.

Scales, especially in the enlarged maps would be better being scale bars.

Figure 2: Simple station names would be preferable here (see comments about Table 1). The labels/legend for the nMDS plot is also very confusing. Please don’t mix colour with symbol: for example, the controls are black, but the legend for site (C15, Oden and Control) is also black, which is confusing. Just one legend would suffice, with each symbol (unique in both color and shape, e.g, purple circle vs red circle) referring to one category. Also, please remove the boxes Primer puts in automatically (such as the SIMPROF box with a green line, the box in the top left with the basics listed: all these can be mentioned in caption and the text).

Figure 4: Personally, I think that the data in Figure 4 would be better presented as a table.

Experimental design

This is original primary research that falls within the scope of the journal.

Research questions:

The authors do a good job of stating their research questions. Both in the introduction and in the discussion, it is quite clear what the focus of the study is. And it definitely fills an important gap. Arctic seeps in general have not been extensively studied and shallow water Arctic seeps in particular, are conspicuously absent in the literature. The context as well, of comparing shallow water Arctic seeps to the background benthos, and the main results, of finding significant differences in the benthos in response to the presence of seeps is important, making this study a well needed addition to the scientific literature.

Methods:

Overall, the methods are appropriate and adequately described. I have some suggestions though, that are detailed below:

Abstract and first line of Methods: Instead of saying ‘samples were taken……’, state immediately what was done (e.g., grabs and trawls were taken in order to examine macrofauna composition). Similarly, in line 84, please change ‘material was obtained….’ to be more specific.

Lines 85-86 and Table 1: The station names are very confusing. First, background 1-3 is difficult to comprehend, since it is at the C15 seep site and not at a control site. The text here does not mention background sites, it simply says that 12 grabs were done at C15, which appears to include these ‘background’ ones.

Ships and cruises have station names that follow certain nomenclature, but it is not necessary to have those in the manuscript. I suggest simplifying the station names. For example, the site name, with the year, followed by consecutive numbering could be an option (e..g C15_2015-1): At least it is important to use names that immediately make it clear which site is being discussed- it is tough to keep track of the fact that 5947 is C15 but 5953 is Oden.

In the text, it should probably be clearly stated that C15 only was sampled in 2015, and that a different grab was used in this year compared to the other two years of sampling.
Line 101 and 104: please change neutralized formalin to buffered formalin. Please check, but should it be 10% buffered formalin (vs 4% formaldehyde)?

In the Methods overall, regarding the analyses done, the text skips from what was done with grabs and what was done with trawls, which is very difficult to keep up with. I suggest first writing out what was done with grabs and then in a separate paragraph, describing the analyses conducted on trawl samples.

There are some language issues: change ‘weighted’ to ‘weighed’, line 119: density and biomass were scaled up to per square meter, line 118-119: square root transformation does not ‘increase the role of rare taxa’, but it does compress larger values more than smaller values, etc.

Validity of the findings

The statistics and analyses overall are appropriate and I don’t think need to be modified as such. However, I do recommend certain modifications to presentation and discussion of results and analyses.

I don’t understand what C15-seep refers to, since the C15 stations are also seep stations. Understanding this would probably help towards understanding why seep 1 and seep 2 so different from other seep sites, including other C15 seep sites. Is it simply that these two sites form their own cluster?

The authors discuss, in detail, the five clusters that they display in Figure 2. I understand that these clusters showed up at the level of similarity shown. But it would be better to approach the whole clustering aspect with some hypotheses. After all, any number of clusters can be ‘made’ based on the similarity index in question. It is more useful to think first, of possible, or expected clusters and then see if the data follows expected patterns or not. For example, one could expect seep sites, control sites and seep background sites to form distinct and significant clusters. Perhaps one would expect the two seep sites to cluster separately as well. Whatever the case, there should be some justification for the clusters being examined, rather than the fact that they simply showed up. And then one can discuss aspects of those clusters such as dominant species, more or less evenness, etc.

In short, I recommend modifying this aspect of the manuscript to align with specific questions and hypotheses. This is particularly in light of the fact that it appears that the authors do have hypotheses in mind, for example, I think they expected seep sites to be quite different from, and cluster separately, from control sites. So please include this and reshape the discussion and presentation of the cluster analysis and nMDS to reflect this. Please note, this does not mean that you cannot discuss the 5 clusters that are currently discussed: it is absolutely acceptable to discuss expected results and then discuss the obtained results, for example, in this case, you would say that in addition to the expected clusters, an additional cluster was the two C15 sites that did not cluster with other C15 seep sites. Furthermore, discussing the cluster analyses and results within the context of expectations and hypotheses would introduce the idea of seep background sites early on, which is needed. It is not clear whatsoever what seep background refers to (as opposed to control sites) and I assume the authors mean something along the lines of areas peripheral, or on the edges of seep locations. But this needs to be explained early on. Also, it should be noted that the background sites are all from C15, ie, Oden has no background sites.

Table 2 and the Kurskal-Wallis tests for testing differences in abundances of taxa: I am not sure I completely agree with using this. I think it is more important to look at community differences and various aspects of the communities could contribute towards one community being significantly different from another, such as species richness, evenness, abundance of certain taxa, etc. I think this is covered in the cluster analysis and nMDS, and Figure 4 even lists numerous aspects by which the communities differ. So then checking if abundances of certain taxa are significantly different seems redundant and also putting an inordinate amount of emphasis on one aspect that contributes towards community differences.

Please change species number throughout to either species richness or number of species.

The use of ‘non randomly’ is not appropriate and not clear in meaning.

Since trawls were only taken at the seep sites and that too, just one each, instead of concentrating on comparing grab and trawl samples as the main focus of the trawl samples, I suggest using the trawl samples to provide more insight into seep communities. Some of these discussions can overlap, for example, the fact that dominant species are different based on grab vs trawl samples. But the main value of the trawl samples is not to simply compare with grabs, in fact, that is not even really appropriate given that only 2 trawls were conducted. But rather, the value of the trawls is that they provide another view, for example, they provide data over more continuous area while grabs are more discrete sampling units. Based on the grab data, it was determined that different communities exist, even the two seep sites are significantly different from each other. The trawls can be used to examine inter-seep differences (or similarities) further. Presenting the trawl data only as a comparison to grab data is, I think, not completely appropriate and downplays the value of this data.

Discussion:

Similar to what I have noted for the cluster analysis, the Discussion would benefit from first stating hypotheses or expectations and then discussing the data in light of these hypotheses. Overall, my impression of the Discussion is that good ideas are introduced, but are not developed well and points don’t always flow well from one to another.
The discussion of increased biomass and abundance/density is confusing the way it is split up and in two parts (paragraphs 1 and 3 of the discussion). I understand that in one case, it is overall abundance and biomass and in the second case, you are referring to certain common species. But it is confusing the way it is. Also, the higher than background diversity of Oden warrants a discussion, since seeps and vents usually are seen as having lower than background diversity.

The section ‘taxa responsible for differences’ ends up being more of a discussion about common shelf species being present at the seep site. The heading should probably be modified to reflect this. In fact, this is something that should be developed further, ie, the fact that Arctic seeps host regular benthic species, albeit in higher numbers and even sometimes displaying higher diversity than the surroundings. Your findings therefore correspond with data from other Arctic seeps that indicate that Arctic seeps display a community structure quite different from seeps in lower latitudes and perhaps these ideas could be developed further here. However, later, there is a sentence saying that the seeps are defined by animals that are obligate for reduced conditions (even though only siboglinids are really obligate in that sense, though Decker and Olu have hypothesized that rissoid snails might be seep specific as well).

In the comparison between seeps and control sites, one important result that is not discussed, but really quite key, is that the control sites are more even, while the seep sites tend to be dominated by certain species. While the increase in biomass and abundance and in the case of Oden, even diversity, is important to discuss about the seep sites, the fact that the non seep sites display higher evenness is also a very important finding that is only mentioned in the results, but not discussed any further.

Note that rissoid snails have been observed also at the Lofoten canyon seep site (Sen et al., 2019 Scientific Reports) and Decker and Olu 2012 (Marine Ecology) have used stable isotopes to determine that rissoid snails at HMMV graze on bacterial mats. Åström et al. are also publishing isotope signatures of Hyalogyrina snails that also seem to be grazers of bacterial mats at Barents Sea seeps (Marine Ecology Progress Series, in press). It would be good to have an image of at least shells of Frigidalvania found in this study, since it is a new species. The authors mention that it was not seen in video imagery. Could this be due to the size? Rissoid snails at Norwegian Sea seeps are mm sized and are only distinguishable in images if the zoom is turned on.

The trawl data could be used to enhance the discussion on a higher variety of microniches at seeps. For example, since trawls are over larger distances, but grabs were taken near flares, it makes sense that grabs were dominated by Oligobrachia, whereas trawls were dominated by other species. Overall, this discussion needs more input: the provision of more habitat types at seep systems as an explanatory factor for the aggregating effect of seeps, including at high latitude seeps, has been discussed in a number of articles, but they are not referred to here. Neither are some of the niches, such as tubes of siboglinid tubes, bacterial mats and probably most importantly, carbonate rocks formed due to AOM. In fact, carbonates are not mentioned at all. Were they not present at all at the sites?

I like the hypothesis put forward by the authors, about oligotrophic conditions selecting for chemosynthesis based primary production at the shallow depths of the study sites. It is a pity though, that this discussion is so limited and resigned to the very end, almost as an afterthought. Is there any data to support their hypothesis, e.g., sediment organic carbon content at the study sites, seasonal data, etc.?

One point of discussion notably absent is a comparison between the two seep sites. The authors found that though the seep sites clustered together and separately from the control sites, the two seeps themselves formed distinct clusters. And even the trawl data revealed differences in the communities between the two seeps. Is there any environmental data that could contribute towards discussing the differences between the two seeps? Neither porewater chemistry nor sediment pigment data is presented, and bathymetry as well as information on carbonate concretions is also absent. It is hard to believe that no such information at all was collected. Not just for comparisons between the seep sites, but comparing between seep vs control and/or ‘background’ sites should also include information on environmental factors. The manuscript as a whole would greatly benefit from including some environmental information, since seep communities are so strongly determined by local conditions.

Additional comments

Thank you for publishing data on shallow seeps in the Laptev Sea. This will be a good addition to the scientific literature overall, though currently, some changes to the discussion are needed.

Reviewer 2 ·

Basic reporting

This is a self-contained manuscript, relatively easy to read and understand. It adds to our knowledge of the distribution of species associated with seeps, particularly in shallow waters of the Laptev Sea. References are up-to-date, background and context are appropriate. Figures are relevant but need attention to some details. Raw data (taxa, abundance, biomass) is provided in a Supplementary Document, which this reviewer has not seen. It would be good to confirm that this data is provided for each sample and each trawl.

Figure 1. These maps do not show the location of the methane plumes for each site, nor does it show bathymetric data for context. In addition, it would be good to include scale bars – Lat/Lon is fine, but the length of 1 km, for example, would make it easy for the reader to understand in a glance the distances between sites (and relative to known seep flares). What does AMK stand for?

Figure 2. Apparent ambiguity: Is A also biomass data square root transformed? Or just B as written. If just B, what data was used for A? Are bot A and B needed? Or is the info redundant?

Figure 4. Should the order be Control, Seep Background, C15, Oden (seep?), C15Seep? I.e., place like next to like and provide more descriptive titles? Should Seep Background really be “Control” as well? What about C15? Without some rules for how sites are named, it is hard for a reader to see at a glance what the authors mean to convey.

Lines 142-147: Seems an odd summary, since the focus of the study is on seep vs non-seep fauna. What do the authors expect the reader to take away from this paragraph? Impossible to contrast seep/non-seep from this.

Line 156: correspondence with presence/absence of seeps – but without showing seep locations on Fig 1, this is impossible for the reader to evaluate.

Line 169: this reder remains confused about whether the C15 station group is seep or background(control).

Line 198: should read “comparison of seep and non-seep station groups”

Line 254: Confusing first paragraph – now the focus is on stations, not station groups. “All gears” refers to two types of grabs and the trawl?

Line 270: Again, the focus is on stations, not station groups, yet we have learned that there are clusters/groups. Why are they put back together again here? This does not seem analytically/ecologically sounds\

Experimental design

The experimental design includes 2 trawls across seep environments. Given that seeps are often recognized as vulnerable and colonized by long-lived, endemic taxa, it might be helpful to provide a justification for trawling. This reviewer recognizes that this could be a sensitive point, but felt it must be made; in this reviewer’s opinion, trawling as part of an experimental design for seep studies should be discouraged, unless an environmental impact statement makes it clear that there will likely be no serious harm. A survey with remote-sensing instruments (i.e., cameras) would have been ideal as a means to assess megafaunal diversity and distribution, rather than destructive trawling. I think for this work, it is important to include the trawl data, but perhaps the authors could craft a way to indicate this is a less than ideal approach for assessing sensitive deep-sea habitats (with a brief explanation), with a note that in this case, it was expedient given tools available and that elsewhere seeps at such shallow depths do not support endemic taxa? The point I hope would be conveyed is that trawling is not what the deep-sea scientific community (or at least, many members of it) considers to be best scientific practice.

There is a bit of circular logic, defining seep sites by location and then defining seep sites by the faunal assemblage. The reviewer understands what was done, but perhaps the authors should be explicit in the methods about how they will define a seep based on the fauna?

Even anecdotal evidence of differences or similarities of seabed substrata among sites would be useful. Were there authigenic carbonates at the seep sites? Any differences in sediment quality?

There seem to be some imprecision and over-statement. Two examples from the abstract:
“The aim of this work was to understand the effect of seeping methane on benthic macrofauna at depths 60-70 m.”
The work does not look at the effect of seeping methane on benthic macrofauna. It does provide descriptive measures of faunal assemblages, some of which are located in close proximity to methane flares, from which association with methane seeps is inferred, but no measure of methane is provided, no direct association between macrofauna and methane concentration is reported. And it may be that sulfide, which is typically co-produced with methane may have as much or more of an ‘effect’ in structuring benthic communities. It is impossible to say from this data, though one may speculate.

“The development of specific methane seep communities at such a shallow depth apparently is related to pronounced oligotrophic environment on the northern Siberian shelf.”
Again, there is no experimental evidence of this; the authors do write “apparently”, but perhaps could be more explicit by expressing this as an hypothesis.

This reviewer has not scoured the ms looking for other ‘over-statements’. These are not unusual to discover in initial submissions, but the authors would do well to look for these and turn any remaining unjustified statements into hypotheses and discussion of alternatives.

A key concern, as mentioned above, is how stations and station groups are analyzed either as sites or groups and then interpreted. There is a confounding of habitat types when the analysis is by site, as the authors show very well.

Validity of the findings

Given the finding of groups, the validity of site analyses is in question.

And as also noted above, some hypotheses are presented as fact, which is not valid.

Reviewer 3 ·

Basic reporting

see below

Experimental design

see below

Validity of the findings

see below

Additional comments

General: This is an interesting study that expands our knowledge of methane seeps to the shallow waters of the Siberian Arctic. The data are appropriately analyzed and conservatively interpreted. I have a number of relatively minor concerns that I itemize below. I believe, however, that the manuscript could be more clearly presented and have a greater impact than in its current draft. The manuscript needs to be edited by an English speaker to improve the grammar, syntax, and word choice. There are numerous grammatical errors and awkward sentences. I also found it very hard to follow because it was not always clear what was a seep community and what was reference. It is important to identify sites by station number so they can be cross referenced with other studies, but for a reader unfamiliar with the stations, cruises, etc. it makes it very difficult to follow and interpret the results. As I point out below, it is also important to show the spatial relationship between the samples and the flares.
Introduction
There needs to be more background on seeps in general. The introduction assumes the reader is well acquainted with seeps in the Arctic and at lower latitudes. One or two paragraphs should be sufficient. And there needs to be more review of the input of phytoplankton vs. methane derived carbon to seeps and the relationship to depth as that relates to the conclusions (expand line 49).
There also need to be more details on the Laptev Sea seeps. What are the characteristic features of the seep field (l 64-65)?
More details are need on the earlier cruises (l 69-72) or omit this section.
Methods
You indicate that seeps were identified by acoustic methane flares. It would be very helpful to indicate the relationship between the samples and the flares. You report that several grabs were off the mark as determined by their contents (which is circular reasoning). A map of grabs relative to the flares is needed.
What is the relationship between the trawls and seep sites. The maps need to clearly identify these relationships. I don’t think the trawl data contribute that much to the story, but frankly it is hard to tell because I don’t see their spatial relationship to the seeps.
L 85. This is where the description of the sample sites needs more clarification. Identifying sites as C15 and Oden are fine, but each sample needs to be identified as seep or non-seep.
L 47 Some of the sites from Åström et al., 2016 are less than 90 m.
L 100 ‘most’ grabs. Not all?
L 111 More detail on these indices. What H’- log10 or log2 or ln?
Results
L 156 More detail is needed here to help the reader see the patterns you are talking about.
Figure 1 As noted, the description of stations and their relationship to the seeps is very hard to follow. Clearly identify what is seep and what is background/reference. C15 has both. Which is station 5625 for example? A reader needs to be able to get all they need from a figure and legend without refereeing to the text or table.
Figure 6 is not useful.
Table 2. The means of each of the stations need to be reported and then the statistical results of the KW tests can be reported in one column. The current format makes it very hard to see the results and impossible to see the magnitude of the differences.
Discussion
In general, there needs to be more discussion of the results in the context of other methane seep studies, Arctic and non-Arctic. The discussion is very narrowly focused and does not relate the patterns in the Laptev to other seep areas.
L 345 But Åström et al., 2018 used a 0.5 mm mesh. Some speculation on why none of the Arctic seeps have large bivalves would be welcome.
L 364 More discussion is needed on the importance of habitat. Carbonate precipitates as a result of anerobic methane oxidation and this enhances hard substrate and habitat complexity. Work in the Arctic and elsewhere has addressed this. The relationship between the n-MDS results and the microhabitats needs more support. You refer to ROV work in the area. What do images reveal about the habitat?
L 373 This is the only paragraph that really tries to interpret the results in a larger context. How do the primary productivity values compare to other Arctic seep areas? Is there a continuum? Are you suggesting you only get a methane response in shallow water in oligotrophic areas? What if there is strong pelagic-benthic coupling in deeper areas, would you not expect a response? That is an interesting hypothesis that needs to be developed more fully even if you can’t completely address it with your results. Maybe a table comparing seeps, depths, water column productivity, etc. would be instructive. I can even see a conceptual model; some similar ones exist for vents.
You also need to discuss the differences within a site but between station types (Figure 2). There are seep effects and there are other spatial patterns, you neglect the latter.

---

## Round 0.2 · Major Revisions

I apologise for the long time in getting this MS back to you. This is because I invited 15 referees of whom 13 declined and 2 never replied. Thus I went through the MS myself. I found numerous corrections to the English necessary (see MS Word file) and also have several questions about the methods and analyses. I think the paper would be greatly improved by removing extraneous analyses redundant diversity indices) and better focusing the Results and Discussion on the key message (as nicely summarise in the Conclusions).

---

## Round 0.3 · Minor Revisions

Both referees note that the MS still needs significant improvement to the English. I am sorry but this is not something either the referees of editors or journal can do for you. Reviewer 3 also notes some important points that must be clarified. Please take your time to be much more rigorous in improving the quality of the English in the next version. This has been poor to date and makes the reviewer's work more difficult.

Reviewer 3 ·

Basic reporting

see below

Experimental design

see below

Validity of the findings

see below

Additional comments

The authors have done a good job revising the manuscript and responding to the reviewers’ comments. I still have a few issues with the manuscript, which I outline below. These are relatively minor and though I feel they should be addressed should not prevent the manuscript from being accepted.
1. The language is much improved, but is still in need of an English speaker editor. The following are some of the lines where the syntax, grammar, word choice, or preposition is wrong: 15 (awkward phrasing), 28 (among not between),34 (awkward), 37 (wording), 42 (same as line 15), 44 (certain increase??), 69 (for is wrong preposition), 74 (awkward), 86 (conduct not conduit), 90 (awkward), 107 (gear not gears, or better yet methods), 179 (wide range is awkward), 243 (densities), 259 (elsewhere not randomly???), 339 (not statically reliable? Not significant?), 498 (short arctic months is awkward).
2. Line 157. Dunn’s test usually follows a Kruskal-Wallis why Tukey? Corrected for what?
3. Line 247 Some place the total abundance needs to be reported and included in statistical tests/\
4. L 472 Need to cite: Åström, K.K.L., Carroll, M.L., Sen, A., Niemann, H., Ambrose, W.G. Jr., Lehmann, M.F., Carroll, J. Chemosynthesis influences food web and community structure in high-Arctic benthos. Marine Ecology Progress Series 629:19-42. doi.org/10.3354/meps13101
5. L 491 see papers by Åström. This is the key issue with the paper. You cannot separate the impact of the carbon source and habitat heterogeneity. Few can in this system. But, you need to be more explicit about the confounding issue.
6. Your main point is lost. You need to draw a stronger comparison between oligotrophic and more productive areas. At least make it a separate concluding paragraph. There has been work done in a few other shallow water areas (Argentina for example; M.E. Bravo; mebravo@iado-conicet.gob.ar), that should be cited or referred to.
7. I do not see the point of Figure 6. You cover the material in the text. A table with the percentages of the top 10 species would be more informative than the figure, which has very limited value. Can you make some predictions about other shallow areas?

Reviewer 4 ·

Basic reporting

Update and check suggested references and add references to support your statements in the article.
I recommend and overall check of the English language. Some parts of the text is confusing (eventually because of insufficient English?) and needs clarification.

Experimental design

No major comments.
See details for the attached document.

Validity of the findings

Authors provide support for their concluding findings. Community descriptions of fauna associated with shallow seeps in the Laptev Sea. Seeps are different from non-seeps and they highlight the distinct species Oligobrachia sp.

Additional comments

This is an interesting study because of its location (Arctic Siberian shelf) of these special type of habitats (cold seeps) and the depth of the study site. The study adds in new information of benthic communities at Arctic methane seeps. It is a strength that both seep and non-seep areas are assessed. I find the manuscript in need of both some general improvements and edits (grammar and minor figure edits). I believe, however, that the manuscript would improve by adding a short section or comment on pelagic-benthic coupling or productivity to match the discussion and links to the shallow seep-water systems and especially to the comments/statement in the last section of the discussion.

Annotated reviews are not available for download in order to protect the identity of reviewers who chose to remain anonymous.

---

## Round 0.4 · accepted · Accept

Thank you for the significant and thorough improvements to the language and paper.